# GTPase-activating protein Rasal1 associates with ZAP-70 of the TCR and negatively regulates T-cell tumor immunity

Youg Raj Thaker[1,2], Monika Raab[3], Klaus Strebhardt [3] & Christopher E. Rudd[1,4,5]*

Immunotherapy involving checkpoint blockades of inhibitory co-receptors is effective in combating cancer. Despite this, the full range of mediators that inhibit T-cell activation and influence anti-tumor immunity is unclear. Here, we identify the GTPase-activating protein (GAP) Rasal1 as a novel TCR-ZAP-70 binding protein that negatively regulates T-cell activation and tumor immunity. Rasal1 inhibits via two pathways, the binding and inhibition of the kinase domain of ZAP-70, and GAP inhibition of the p21$^{ras}$-ERK pathway. It is expressed in activated CD4 + and CD8 + T-cells, and inhibits CD4 + T-cell responses to antigenic peptides presented by dendritic cells as well as CD4 + T-cell responses to peptide antigens in vivo. Furthermore, siRNA reduction of Rasal1 expression in T-cells shrinks B16 melanoma and EL-4 lymphoma tumors, concurrent with an increase in CD8 + tumor-infiltrating T-cells expressing granzyme B and interferon γ-1. Our findings identify ZAP-70-associated Rasal1 as a new negative regulator of T-cell activation and tumor immunity.

[1] Cell Signalling Section, Department of Pathology, University of Cambridge, Tennis Court Road, Cambridge CB2 1QP, UK. [2] School of Biological Science, Protein Structure and Disease Mechanisms, University of Essex, Wivenhoe Park, Colchester CO4 3SQ, UK. [3] Department of Obstetrics and Gynaecology, School of Medicine, J.W. Goethe-University, Theodor-Stern-Kai 7, 60590 Frankfurt, Germany. [4] Département de Immunologie-Oncologie, Centre de Recherche Hôpital Maisonneuve-Rosemont, Montreal, QC H1T 2M4, Canada. [5] Département de Medicine, Université de Montréal, Montreal, QC H3C 3J7, Canada. *email: christopher.e.rudd@umontreal.ca

The advent of immunotherapy has ushered in a new era of therapies for cancer[1]. Present approaches involve the use of immune checkpoint blockade against programmed cell death 1 (PD-1) and other inhibitory receptors (IRs) to enhance T-cell responses[2,3]. An alternate approach is to directly target negative signaling pathways that limit T-cell responses to neo-antigens in tumors. T-cell activation is mediated by protein kinases p56lck and zeta chain-associated protein kinase 70 (ZAP-70)[4,5]. CD4 and CD8 bind to p56lck [6–8] where the kinase phosphorylates various substrates that include immunoreceptor tyrosine-based activation motifs (ITAMs) on the zeta and CD3 chains of the T-cell receptor (TCR)[9] for the recruitment of the second kinase, ZAP-70[10]. Ligation of the TCR complex, in turn, induces a tyrosine phosphorylation cascade involving various enzymes and adaptor proteins such as linker for the activation of T cells (LAT)[5,11]. The adaptor complex, LAT-SLP-76 regulates calcium mobilization[11–13], while the adaptor SKAP1 (aka SKAP-55) acts a scaffold for the activation of LFA-1[14–16] via the regulation of RapL–Rap1 complex formation[17,18]. SKAP1 also binds to the Polo-like kinase 1 (PLK1) for the optimal cell cycling of T cells[19] and the p21 activating exchange factor RasGRP1[17,20].

The outcome of the activation process also involves inhibitory co-receptors and negative intracellular mediators that prevent excess inflammation and safeguard against the development of autoimmunity. Key among these are programmed cell death protein-1 (PD-1), T-cell immunoreceptor with Ig and ITIM domains (TIGIT), and others which possess immunoreceptor tyrosine-based inhibition motifs (ITIMs) or immunoreceptor tyrosine-based switch motifs (ITSMs) that bind the phosphatases Src homology region 2 domain-containing phosphatases 1 and 2 (SHP-1 and SHP-2). In the case of PD-1, negative signaling has been attributed to SHP-2 binding to the ITSM[2,21,22], where it inhibits the phosphoinositide 3-kinase (PI 3 K)-protein kinase B (PKB) (also AKT) pathway leading to altered metabolism and a block in cell cycle progression[23]. Other negative regulators include the E3 ligases of the ubiquitin pathway[24–26].

The TCR complex also stimulates the small GTPase p21ras that operates upstream of the Ras–Raf–mitogen-activated protein kinase kinase (MEK)/extracellular signal-regulated kinase (ERK) signaling pathway[27]. p21ras is active when bound to GTP, and inactive in a GDP form. Guanine nucleotide exchange factor (RasGEF) RasGRP1 and Son of Sevenless (SOS) stimulate the release of GDP in exchange for GTP. GRB-2-associated SOS binds the YMNM motif of CD28[28,29] and the LAT complex[30]. Conversely, GTPase-activating proteins (RasGAPs) inhibit p21ras by promoting conversion of active GTP-bound Ras to the inactive GDP-bound form[31]. The inactivation of the p21ras-ERK pathways has been linked to the induction of T-cell nonresponsiveness or anergy[32,33].

The Ras protein activator-like 1 protein (Rasal1) is a member of a GAP family that includes p120 RasGAP (or Rasa1), NF1[34,35], and Rasa2-4[36–38]. Each contains two N-terminal C2 domains which is followed by a GAP domain, a PH domain, and a Bruton tyrosine kinase (BTK) motif[39]. The PH domain of Rasa2 and Rasa3 mediates membrane recruitment, while the Rasal1 C2 domain senses levels of intracellular calcium[36]. By sensing oscillations in intracellular calcium, Rasal1 shuttles between the membrane and cytoplasm. Translocation to the membranes is driven via the C2 domain binding the membrane phospholipids. In addition to p21ras, Rasal1 shows GAP activity toward the Ras-related small GTPase, Rap[40]. Purified Rasal1 alone is devoid of RasGAP activity since it requires a change in GAP domain confirmation as triggered by membrane localization. The calcium-dependent C2 domain interaction with membrane phosphatidylserine and phosphatidylinositol lipids is the critical event that triggers its RasGAP activity[41].

Altered or mutated Rasal1 has also been linked to various cancers[42–44], where it serves as a tumor suppressor in human thyroid[45], colon[44], liver[46], and thyroid cancers[47].

Despite its importance in the transformation of cells, the role of Rasal1 in T-cell activation and cancer immunotherapy has yet to be explored. Here, we have shown by a combination of tandem affinity chromatography, co-precipitation, and proximity ligation analysis (PLA) that Rasal1 binds to the kinase domain of ZAP-70 of the TCR complex and inhibits anti-CD3 activation of ERKs in T cells. It regulates in vitro and in vivo T-cell responses to peptide antigen. Further, the knockdown of Rasal1 in T-cells limits B16 melanoma pulmonary metastasis and the growth of solid EL-4 lymphoma tumors. This enhanced anti-tumor effect is accompanied by a marked increase in CD8 + tumor-infiltrating T-cells (TILs) expressing killer effector molecules granzyme B (GZMB) and interferon γ-1 (IFNγ1). Overall, our findings identify a novel TCR-associated protein that appears after T-cell activation and negatively regulates T-cell activation and antitumor immunity.

## Results

**Rasal1 is a TCR-associated GAP in T cells**. TCR zeta tandem affinity purification (TAP) and mass spectrometry (MS) were used to identify novel TCR/CD3 interaction partners in reconstituted TCR-null zeta[−/−] 3A9 cells[48,49]. The TCR-null zeta[−/−] 3A9 cell is a murine T-cell hybridoma which was reconstituted with expression of a zeta chain coupled to Strep-II and His/V5 tags (i.e., HIS-STREP-II-tagged zeta) for purification (Supplementary Fig. 1). The sequential two-step purification approach using Strep-II and V5/His tags minimizes the presence of non-specific contaminating material. While TCR-null zeta[−/−] 3A9 cells do not express the surface forms of the TCR, expression of the zeta chain by transfection with our TAP-zeta constructs restored TCR expression (Supplementary Fig. 1). Reconstituted clones with levels of TCR surface expression comparable to peripheral T cells (i.e., Clone 1.2) (lower panel) were selected. Clone 1.2 was sorted by FACS and grown to establish a cell line. The hybridoma clone was then activated for 5 min with soluble anti-CD3 prior to two-step TAP sequential chromatography to co-purify associated proteins for analysis by SDS-PAGE and silver staining, as well as tandem mass spectrometry (MS/MS). From this, MS/MS identified peptides for the T-cell receptor alpha, beta, gamma chains, and CD3 and zeta subunits (Fig. 1a, right panel). Under the same conditions, four peptides corresponding to a Ras GTPase-activating like 1 (Rasal1) were detected in each experiment ($n = 3$) (Supplementary Fig. 2). A TCR-specific protein at 90–95 KD, the Mr for Rasal1 was also seen by silver staining of SDS-PAGE (left panel). For technical reasons involving the need to cut the gel for MS/MS analysis, a gel slice in the region of 70 kDa was excluded, and hence, ZAP-70 peptides were not seen by MS/MS. However, confirmation of the presence of ZAP-70 in our 3A9 cells was seen by anti-CD3 precipitation followed by anti-ZAP-70 blotting (Fig. 1b, lanes 1 and 2). Furthermore, the level of co-precipitated ZAP-70 increased following anti-CD3 stimulation for 10 min (lane 2 vs 1). Overall, using TAP analysis, these studies showed that Rasal1 associates with the TCR–CD3 complex in T cells.

To assess Rasal1 expression in T cells, murine naive CD4 and CD8 + T cells were ligated with anti-CD3 antibodies followed by a measure of mRNA levels for Rasal1 and related Rasal3 (Fig. 1c). CD4 + T cells showed upregulation of both Rasal1 and 3 by 24 h followed by a reduction in their expression at 48 h. By contrast, CD8 + T cells showed a sustained increase in the expression of Rasal1 at 24 and 48 h, while Rasal3 did not increase with stimulation. Overall, these data showed that Rasal1 is expressed in CD4 and CD8 + activated T cells in response to TCR ligation.

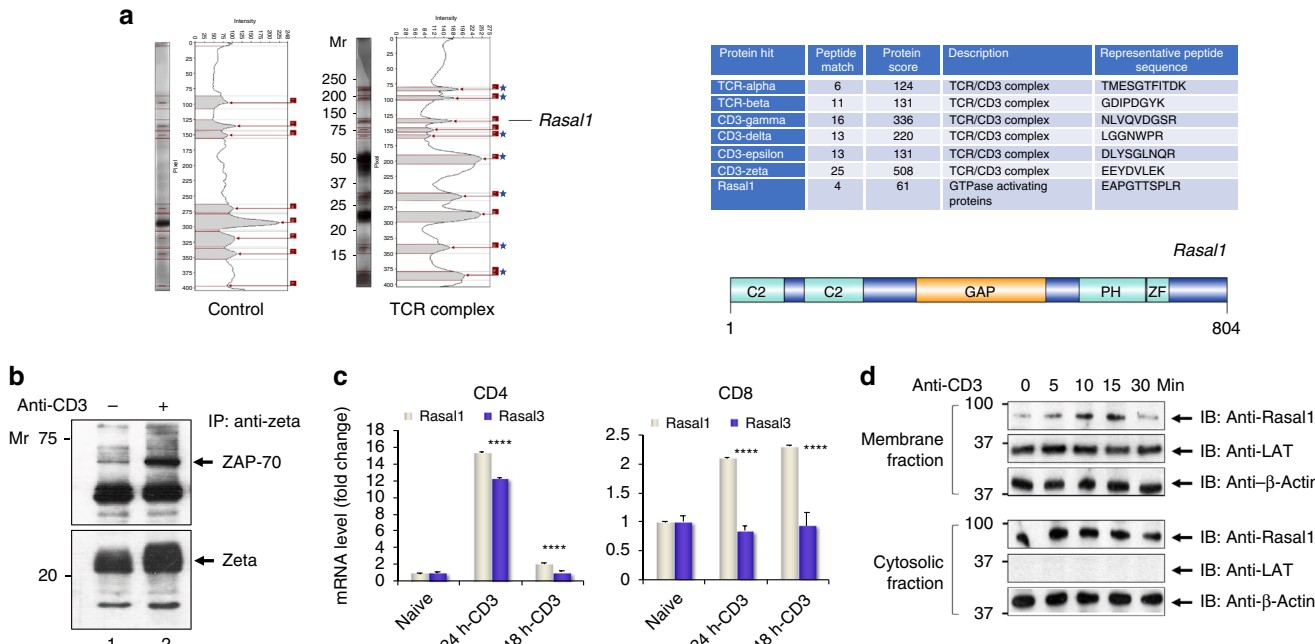

**Fig. 1** TCR complex purification and Rasal1 association with ZAP-70 in the TCR complex. **a** Analytic gel from tandem affinity purification (TAP) of TCR plus associated proteins captured from TCR-constituted cells (TCR complex) versus TCR-null negative control cells (control) (left panel). Chart showing the list of TCR-associated proteins identified by MS analysis (right upper panel). The structure of Rasal1 (right lower panel). **b** Co-precipitation of ZAP-70 with TCR from 3A9 T cells. 3A9 cells were stimulated for 10 min with anti-CD3 followed by immunoprecipitation with anti-TCR zeta and blotting with anti-ZAP-70 or anti-zeta. Lane 1: resting T cells; lane 2: anti-CD3-stimulated T cells. **c** Rasal1 expression is induced with anti-CD3 activation. Relative fold change expression of Rasal1 versus Rasal3 upon anti-CD3 stimulation of CD4$^+$ and CD8 + peripheral T cells. 18 s rRNA served as a control. **d** Anti-CD3 induces Rasal1 translocation to the membranes of T cells. Stimulation-dependent association of Rasal1 with the membrane (upper panel) versus cytosolic Rasal1 (lower panel). Anti-actin served as a loading control. LAT was used as a marker of membrane enrichment fractionation ($n = 3$)

In addition, we found that anti-CD3 induced the translocation of Rasal1 from cytosol to the membranes of activated T cells (Fig. 1d). Mouse primary T cells were pre-activated with anti-CD3 for 36 h to induce Rasal1 expression, followed by a rest for 24 h and re-ligated with soluble anti-CD3 for various times. Anti-CD3 ligation resulted in the presence of Rasal1 in the membranes for 5–15 mins followed by a decrease at 30 min. As a control, the transmembrane protein, LAT was only found in the membrane fraction with no cross-contamination seen in cytosolic fractions. These data showed that Rasal1 can be induced to associate with the membranes of T cells by anti-CD3 ligation.

To assess the proximity of Rasal1 with the TCR, we also carried out a nearest neighbor examination of proteins using proximity ligation analysis (PLA) (Fig. 2a). Antibodies to ZAP-70 and Rasal1 were employed together with isotype-specific antibodies with the Duolink-TM detection system in Jurkat T cells[50,51]. Anti-Rasal1 and anti-ZAP-70 showed a strong positive signal in response to anti-CD3 ligation, while no signal could be detected in resting cells. DAPI was used to stain the nucleus of cells (see blue). While PLA can show nearest neighbor localization, it cannot be used accurately to determine the intracellular localization of the complexes. These data provided evidence that anti-CD3 can induce close proximity between Rasal1 and ZAP-70 in activated T cells.

Furthermore, biochemically, anti-Rasal1 co-precipitated ZAP-70 from primary murine T cells that had been pre-activated to induce Rasal1 followed by a rest for 12 h and then anti-CD3 ligation for 5 min (Fig. 2b). Anti-Rasal1 co-precipitated ZAP-70 from these primary cells ligated with anti-CD3 (lane 4). Little, if any co-precipitated, ZAP-70 was seen in resting cells (lane 3). Similarly, anti-Rasal1 co-precipitated ZAP-70 upon anti-CD3

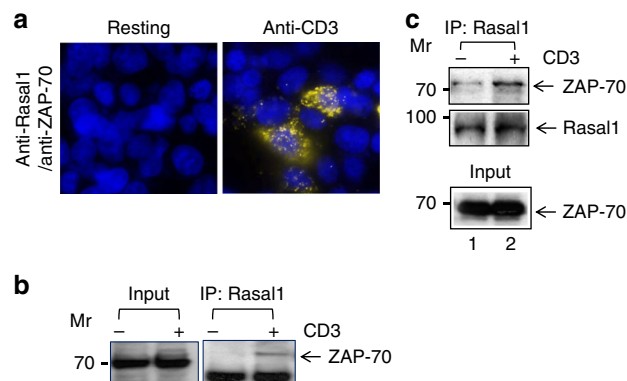

**Fig. 2** Rasal1 binds to the immune cell tyrosine kinase ZAP-70. **a** Proximity ligation assay (PLA) of Rasal1 and ZAP-70 in Jurkat T-cells stained with DAPI, anti-Rasal1, and anti-ZAP-70. Positive PLA signal (fluorescent dots) indicates a distance of 30–40 nm between Rasal1 and ZAP-70 in anti-CD3-activated T cells, indicating their close proximity ($n = 2$). **b** Anti-Rasal1 co-precipitates ZAP-70 from primary mouse T cells. Lane 1–2 loading controls, lanes 3-4: anti-Rasal1 IP: lanes 2 and 4. Cells were stimulated with anti-CD3 ligation for 5 min vs IgG control ligation in lanes 1 and 3. **c** Anti-Rasal1 co-precipitates ZAP-70 from transfected Jurkat T cells. Lanes 1: nonligated (i.e., IgG); lane 2: anti-CD3 ligated T cells for 5 min. Lower panel shows similar level of ZAP-70 expression in lysates of ($n = 2$)

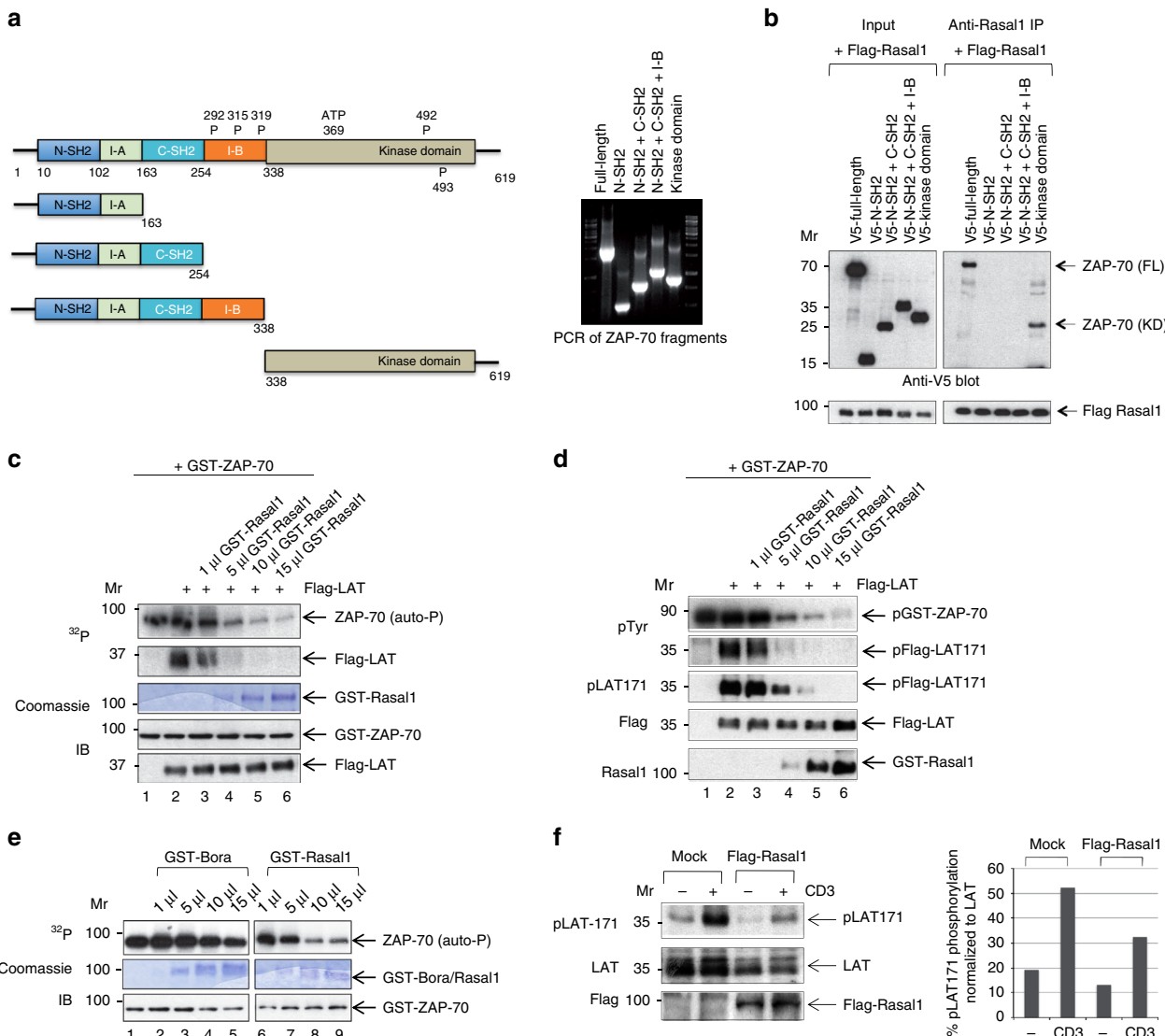

**Fig. 3** Rasal1 binds to the kinase domain of ZAP-70 and inhibits ZAP-70 kinase activity. **a** Deletion mutants of ZAP-70 that include the V5-tagged N-terminal SH2 domain plus the intervening region, the N-terminal SH2 domain plus the intervening region and the C-terminal-oriented SH2 domain, the N-terminal SH2 domain plus the intervening region and the C-terminal-oriented SH2 domain plus the intervening I-B region and the kinase domain of ZAP-70. Right panel: PCR of ZAP-70 fragments. **b** The Rasal1 binds to the kinase domain of ZAP-70. Each of the proteins were expressed in 293T cells in the presence of Flag-tagged Rasal1 (left panel**)**. Anti-Rasal1 precipitation resulted in the co-precipitation of full-length ZAP-70 and the kinase domain of ZAP-70 (right panel) ($n = 2$). **c** Rasal1 inhibits the kinase activity of ZAP-70 and the phosphorylation of LAT. Recombinant active ZAP-70 from baculovirus was incubated with Flag-LAT from 293T cells as a substrate in the presence of increasing concentrations of GST-Rasal1 and γ-32P ATP for 30 min at 37 °C. Increasing concentrations of GST-Rasal1 inhibited ZAP-70 autophosphorylation and the phosphorylation of LAT (lanes 3–6). Lower panels: input levels of GST-ZAP-70 and LAT were assessed by western blotting ($n = 2$). **d** Rasal1 inhibits the kinase activity of ZAP-70 and the phosphorylation of LAT. Recombinant active ZAP-70 from baculovirus was incubated with Flag-LAT from 293T cells as a substrate in the presence of increasing concentrations of GST-Rasal1 and cold ATP for 30 min at 37 °C. Increasing concentrations of GST-Rasal1 inhibited ZAP-70 autophosphorylation and the phosphorylation of LAT ($n = 2$). Lower panels show blots for Flag-LAT and GST-Rasal1 expression. **e** Rasal1, but not Bora, inhibits the kinase activity of ZAP-70 and the phosphorylation of LAT. Recombinant active ZAP-70 from baculovirus was incubated with GST-Bora or GST-Rasal1 with γ-32P ATP for 30 min at 37 °C. Increasing concentrations of GST-Rasal1, but not GST-Bora, inhibited ZAP-70 autophosphorylation. GST-Bora (lanes 2–5); GST-Rasal1 (lanes 6–9). **f** LAT phosphorylation of Y-171 is reduced in the presence of Rasal1 in Jurkat T-cells. Jurkat cells were transfected with Rasal1 and ligated for 5 min with anti-CD3. Rasal1 transfection (lanes 3, 4); resting (anti-IgG) (lanes 1, 3); anti-CD3 (lanes 3, 4) ($n = 3$). Right panel: histogram showing the quantification of phospho-LAT normalized to total LAT levels in the blot

ligation of human Jurkat T cells (Fig. 2c, lane 2). Overall, the combined use of TAP, PLA, and co-precipitation analysis showed that a portion of Rasal1 associates with the ZAP-70 and is recruited to the TCR upon activation.

**Rasal1 binds to the kinase domain of ZAP-70.** In order to assess the site of binding between Rasal1 and ZAP-70, various deletion

mutants of ZAP-70 were cloned and expressed for binding studies (Fig. 3a). These deletion mutants included the N-SH2 domain with the intervening I-A region, the N- and C-SH2 domains with the intervening I-A region, the N- and C-SH2 domains with the intervening I-A and I-B region, and the isolated kinase domain (left panel). The PCR products for the ZAP-70 fragments are also shown (right panel). Each protein was then expressed in

293T cells in the presence of Flag-tagged Rasal1 (Fig. 3b). Intriguingly, anti-Rasal1 co-precipitated only full-length ZAP-70 or the kinase domain of ZAP-70. These results indicated that Rasal1 bound to the kinase domain of ZAP-70.

To assess the implications of binding, we then measured the kinase activity of GST-ZAP-70 in the presence and absence of GST-purified Rasal1 (Fig. 3c–f). We first assessed ZAP-70 autophosphorylation and the phosphorylation of LAT, an established substrate of ZAP-70[52]. Assays were run using radioactive ATP (i.e., hot assay) (Fig. 3c), or as a cold kinase assay (Fig. 3d). Recombinant active ZAP-70 isolated from baculovirus was incubated with Flag-LAT from 293T cells, as a substrate, in the presence of increasing amounts of GST-Rasal1 for 30 min at 37 °C. In the case of the radioactive assay using γ-32P ATP, GST-ZAP-70 alone showed the presence of an autophosphorylation band at 76 KD (Fig. 3c, upper panel, lane 1). The addition of LAT resulted in the labeling of LAT at 40 KD (lane 2), while the titration of increasing amounts of Rasal1 inhibited both autophosphorylation and LAT labeling (lanes 3–6). Commassie staining revealed the presence of exogenously added GST-Rasal1, while immunoblotting confirmed the presence of equal levels of GST-ZAP-70 and Flag-LAT (lower panels).

Similarly, in the presence of cold ATP, immunoblotting with anti-phospho-tyrosine identified a GST-ZAP-70 band (Fig. 3d, upper panel, lane 1), while the addition of Flag-LAT showed the presence of the phosphorylated LAT at tyrosine 171 (Y-171) (lane 2). By contrast, the addition of increasing amounts of GST-Rasal1 markedly diminished the autophosphorylation of ZAP-70 and the presence of phospho-Y-171 of anti-Flag LAT precipitates (lanes 3–6). Anti-LAT confirmed the presence of equal amounts of LAT, while Rasal1 blotting confirmed the increasing amounts of GST-Rasal1 added to the assay (lower panel). GST-Bora (protein aurora borealis), with the same MW as Rasal1, was also used as a control (Fig. 3e). However, the addition of increasing amounts of GST-Bora had no effect on the autophosphorylation of ZAP-70 in a radioactive kinase assay (lanes 1–5). This contrasted with the inhibitory effect of GST-Rasal1 on ZAP-70 autophosphorylation (lanes 6–9). Commassie staining confirmed the presence of exogenously added GST-Bora and GST-Rasal1, while immunoblotting confirmed the presence of equal levels of GST-ZAP-70 (lower panels).

Lastly, we monitored the phosphorylation of the adaptor LAT at Y-171 in Jurkat T cells (Fig. 3f). Jurkat cells were transfected with mock vector (i.e., SRalpha) or Flag-Rasal1 prior to ligation with anti-CD3 and immunoblotting. Anti-CD3 induced the phosphorylation of LAT of Y-171, as previously reported[10,51]. The co-expression of Flag-Rasal1 reduced the phosphorylation of LAT consistent with the notion that Rasal1 inhibits ZAP-70. The normalization of the p-LAT Y171 signal relative to the expression of LAT confirmed that the phosphorylation of LAT at Y-171 was reduced by ~50% (histogram, lower panel). Overall, these data showed that Rasal1 can bind to the kinase domain of ZAP-70, and in the process, inhibit its catalytic activity.

**Rasal1 inhibits ERK activation in T cells**. On another level, Rasal1 is also a member of RasGAP family that includes p120 RasGAP and NF1 which inactivate p21[ras] [34,35]. p21[ras] in turn operates upstream of extracellular-activated kinase (ERK) of the Ras–Raf-MEK-ERK pathway[53]. To investigate this pathway, primary mouse T cells were transfected with Flag-Rasal1, and ligated with anti-CD3 for 5 min followed by blotting against phospho-activation sites Thr202/Tyr204 on ERK (Fig. 4a). Levels of active ERK1,2 were measured by phosphorylation at residues Thr202/Tyr204[54]. While anti-CD3 induced strong pERK1,2

phosphorylation, the expression of transfected Rasal1 inhibited this activation as seen in examples of two representative experiments (upper and lower panels).

Flow cytometry also showed that Rasal1-HA reduced pERK1,2 when gated on CD3 + Rasal1-HA + cells over a time course of 15 min (Fig. 4b, upper panel). Levels of pERK1/2 increased until 10 min followed by a decline at 15 min. By contrast, the transfection and expression of HA-Rasal1 markedly inhibited (>90%) ERK activation at all times. As a control, anti-CD3 induced p38 activation as detected by phospho-p38 MAPK (Thr180/Tyr182) antibody was unaffected by the expression of Rasal1-HA (lower panel).

The phosphorylation of proximal activators of ERK, Raf, and MEK-1 on phospho-activation sites Raf-Ser338 and MEK-Thr286 were also examined (Fig. 4c). In both cases, anti-CD3 was seen to increase phosphorylation in mock transfected cells (lane 2 vs 1). By contrast, the transfection with Flag-Rasal1 reduced the phosphorylation of both mediators (lane 4 vs 2). Collectively, these data showed that Rasal1 inhibits anti-CD3-induced p21[ras]-ERK activation in T cells.

**Rasal1 inhibits T-cell activation**. Given these results, we next assessed the role of Rasal1 in regulating T-cell activation. For this, primary mouse T cells from DO11.10/Rag[2−/−] TCR transgenic mice were transfected with Flag-tagged Rasal1 (Fig. 5a). SRalpha (i.e., mock) and Rasal1- transfected cells were then activated in vitro by bone marrow-derived dendritic cells (BMDCs) exposed to ovalbumin (OVA) peptide antigen followed by an examination of proliferation on day 4. BMDCs were isolated and generated as described[17,55]. Rasal1 expression reduced proliferation by 50–60% at 5 μg and 10 μg OVA as monitored by [3]H-thymidine incorporation. CTLA-4 Ig served as a control to block the response. Blotting confirmed the expression of Rasal1 (upper inset). Likewise, a titration of anti-CD3 showed that Rasal1 inhibited proliferation as monitored by [3]H-thymidine incorporation (Fig. 5b). Inhibition was seen less evident at lower anti-CD3 concentrations of 0.1–0.5 μg/ml, but pronounced at 80–90% relative to vector control when higher concentrations of anti-CD3 from 1.0–6.0 μg/ml were used to stimulate cells. This finding suggests that Rasal1 regulation of T-cell proliferation is affected by the strength of the TCR signal. Furthermore, inhibition at 2 μg/ml was seen at both 48 and 72 h post-anti-CD3 ligation (upper inset). Similarly, expression of Rasal1 inhibited the anti-CD3 increase in NFAT-AP2 IL-2 promoter activity when co-transfected into Jurkat T cells (Fig. 5c). By contrast, esiRNA (enhanced siRNA) knockdown (KD) of endogenous Rasal1 potentiated the proliferation of primary T cells (Fig. 5d). Blotting with anti-Rasal1 confirmed the reduction in Rasal1 expression (right upper panel).

As an additional control, the contact time between T cells and dendritic cells (DCs) was measured using live cell imaging (Fig. 5e). Rasal1 transfection had no effect on contact times unlike in the case of immune cells adaptors, such as ADAP and SKAP[15,17]. These data showed that Rasal1 negatively regulates the proliferation of T cells without affecting the contact times between T cells and presented antigen-presenting cells.

**Rasal1 modulates in vivo responses to antigen**. We next assessed whether Rasal1 could affect in vivo proliferation (Fig. 6). For this, DO11.10 T-cells transfected with scrambled or Rasal1 siRNAs were adoptively transferred into Balb/c mice followed by injection with 50ug OVA[257-264] peptide 24 h later. T cells from lymph nodes were isolated on day 5, and assessed for numbers of cells and activation markers (Fig. 6a). Flow cytometry confirmed the reduction in Rasal1 expression

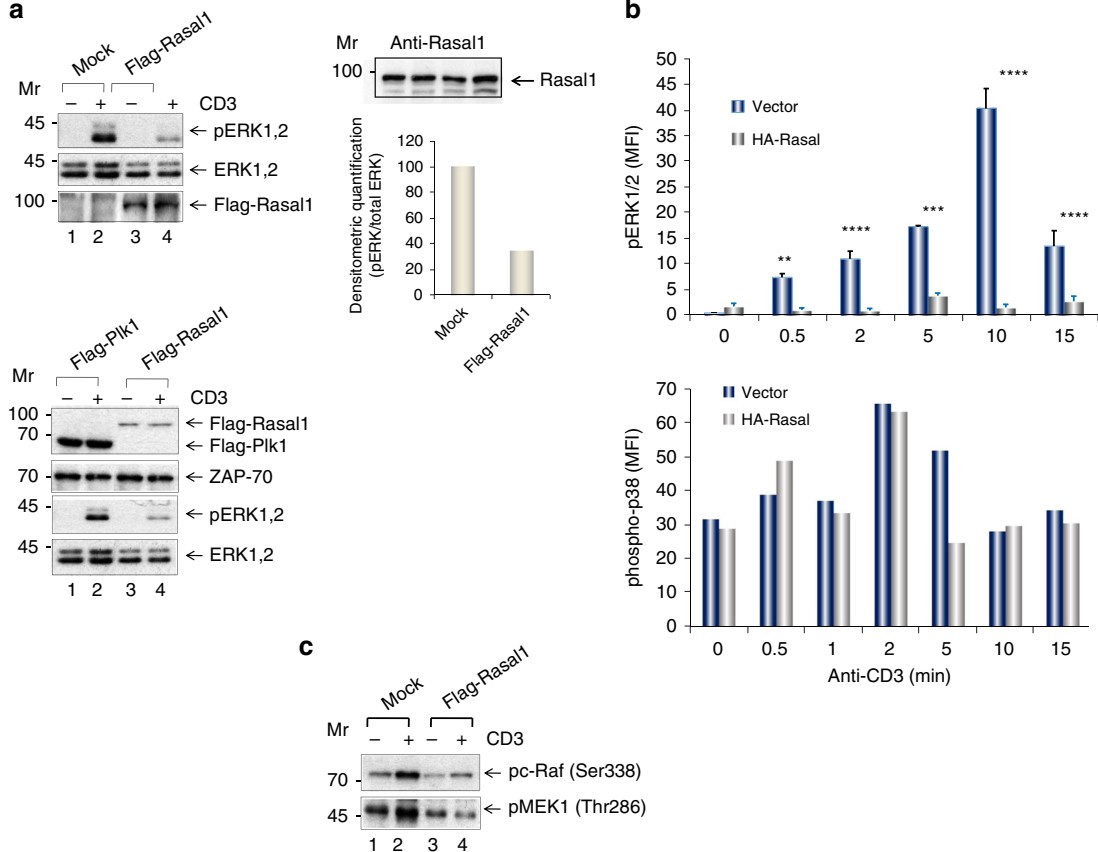

**Fig. 4** Rasal1 inhibits the ERK1/2 pathway, but not p38 pathway. **a** Expression of Rasal1 in T cells inhibits ERK1,2 phosphorylation. Upper and lower panels: two independent experiments where primary mouse T cells were transfected with Flag-tagged Rasal1 and were ligated with anti-CD3 for 5 min followed by blotting with anti-phospho-p44/42 MAPK (Erk1/2) (Thr202/Tyr204); anti-ERK1/2, anti-ZAP-70 or anti-Flag. Flag-PLK expression served as a negative control (top panel) ($n > 2$). Upper right panel shows endogenous levels of Rasal1 in activated primary T cells. Lower right panel: histogram showing the inhibition of pERK relative to ERK expression. **b** Rasal1 expression reduces on ERK, but not p38 activation in activated T cells. Anti-CD3-stimulated (0-15 min) cells were gated on Rasal1-HA$^+$ cells and analyzed. p38 activation analyzed by phospho-p38 staining in response to anti-CD3 stimulation for 0-15 min (lower panel). The data are ± SD and $p$-value calculated by Student's $t$ test ($n > 2$). **c** Rasal1 expression reduces on the phospho-activation of Raf1 and MEK1 in T cells. The phosphorylation of proximal activators of Raf and MEK-1 on phospho-activation sites Raf-Ser338 and MEK-Thr286 were examined. In both cases, anti-CD3 increased phosphorylation in mock transfected cells (lane 2 vs 1). By contrast, the transfection with Flag-Rasal1 reduced the phosphorylation of both mediators (lane 4 vs 3)

resultant from Rasal1 siRNA transfection (Fig. 6b). We observed an increase in lymph node (LN) size or volume from a mean of 10.2 to 13.7 mm$^3$ in mice with adoptively transferred Rasal1 siRNA KD cells relative to transferred cells expressing scrambled siRNA (Fig. 6c). Further, the percent of DO11.10 T cells increased from 20% in control to 70% in Rasal1 KD cells (Fig. 6d). DO11.10 T cells were identified using an anti-mouse TCR DO11.10 antibody (BioLegend)[56]. Among these cells, there was an increase in the percent of cells expressing the activation marker Ki-67 (Fig. 6e). We also observed an increased expression of CD25 and CD69 in Rasal1 KD cells (Fig. 6f). No significant change was seen in the expression of differentiation marker CD62L. Although attempts were made to label cells with CFSE, the in vivo detection of dye was poor (data not shown). These data show that the downregulation of Rasal1 can increase the proliferation and cell expansion capacity of antigen-specific T cells in vivo.

**Rasal1 siRNA in T cells reduces tumor growth**. We next assessed whether Rasal1 represented a potential therapeutic target in anti-tumor immunity. To examine pulmonary metastasis, B16F10 melanoma cells pulsed with OVA peptide were injected

intravenously (i.v.) into C57/BL/6 mice, as we have previously described[57,58]. This was followed by the adoptive cell transfer of purified OT-1 CD8$^+$ T cells into mice (Fig. 7a, upper panel). T cells had been transfected ex vivo 24 h earlier with scrambled (SC) or Rasal1 siRNA. Lungs were harvested on day 14, and pulmonary metastasis was monitored by counting tumor foci ($n = 4$). Consistent with their recognition of OVA peptide, OT-1 T cells expressing scrambled siRNA reduced the number of foci on lungs when compared with control mice that had been injected with PBS alone (lower left panel and right histogram). However, T cells expressing siRNA for Rasal1 significantly reduced the number of B16 foci relative to the action of scrambled OT-1 cells. These data indicated that a reduction in Rasal1 expression enhanced the ability of adoptively transferred T cells to limit the growth of B16 tumors.

We next examined the nature of the tumor-infiltrating lymphocytes (TILs) isolated from the melanoma tumors (Fig. 7b). From this, we observed a major increase in the numbers of CD8 + T cells expressing Rasal1 siRNA relative to CD8 + T cells expressing scrambled siRNA (see left histograms). When averaged over several experiments and expressed as a percentage of the TILs, we observed an increase in CD8 + T cells within the TIL population from 12 to 33% (upper right panel) ($n = 4$).

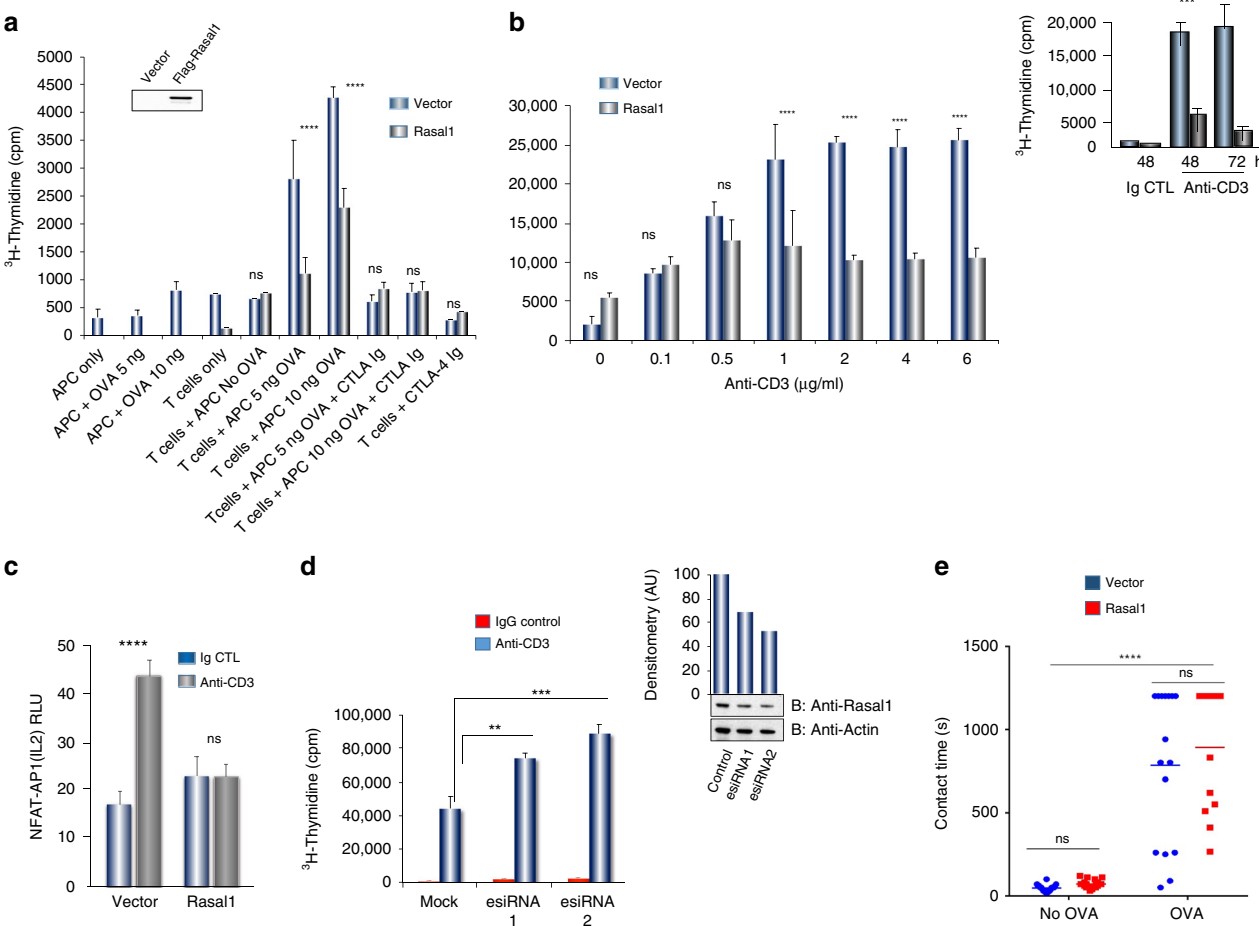

**Fig. 5** Rasal1 inhibits in vitro anti-CD3 and dendritic cell-peptide induced T-cell activation. **a** Antigen (OVA$_{323-39}$)-dependent activation and proliferation of DO11.10 CD4 + cells expressing Flag-tagged Rasal1 compared with control cells at different OVA peptide concentrations (5–10 ng). CTLA-Ig was used as a control. Upper inset shows expression of Flag-tagged Rasal using anti-Flag in blotting ($n = 3$). **b** Inhibition of T-cell responses by Rasal1 to increasing anti-CD3 concentrations. Upper inset: effects of Rasal1 at 48 and 72 h post-anti-CD3 activation ($n = 3$). **c** Anti-CD3-driven NFATc1/AP1 promoter activity in Jurkat T cells was inhibited by the presence of Flag-tagged Rasal1 ($n = 3$). **d** siRNA (esiRNA) knockdown of Rasal1 enhances T-cell proliferation. esiRNA were tested for their knockdown efficiency by western blot (top right panel), esiRNA1 and 2 correspond to 500 nM and 1000 nM concentrations ($n = 3$). **e** Contact time between D011.10 CD4$^+$T-cells expressing Flag-Rasal1 (red) or empty vector (blue) and OVA peptide on bone marrow dendritic cells (BMDCs). Dot blot of T cell/DC contact duration time in presence of OVA was unaffected by Flag-tagged Rasal1 or vector transfection ($n = 3$). Representative plots show mean with ± SD. *p*-value was calculated by unpaired **t** test. *$p < 0.05$, **$p < 0.01$, ***$p < 0.001$

Furthermore, in another experiment which compared CD4 and CD8 cells, we did not observe a change in the percentage of infiltrating CD4$^+$ cells (lower right panel). These data indicated that the enhanced regression of the B16 tumors with reduced Rasal1 expression was accompanied by an increase in the presence of CD8 + TILs.

Furthermore, TILs from tumors of mice injected with T cells expressing Rasal1 siRNA showed a significant increase in the presence of CD8 + cells expressing granzyme B (GZMB) and IFNγ1, the effector molecules of the CD8 + cytotoxic T cells (Fig. 7c, left panels). The percentage of CD8 + cells expressing IL-2 and TNFα showed upward trend, which was not statistically significant (right panels). Overall, using adoptive cell therapy, we found that Rasal1 KD promoted an increase in the presence of CD8 + TILs in tumors with an increased expression of GZMB and IFNγ1 which was accompanied by reduced pulmonary metastasis of B16 tumors in mice.

We next assessed responses to a second tumor model, the EL-4 T-cell solid lymphoma model. Priming of OT-1 OVA-specific T cells with SIINFEKL peptide of OVA (OVA$_{257-264}$) produces a specific cytolytic T-cell (CTL) response against tumor targets[59].

EL4 cells were pre-treated with 2 μg of OVA peptide and washed prior to injection. Tumors were allowed to grow until day 5 followed by injection of scrambled or Rasal1 siRNA expressing cells ($2 \times 10^5$) for 10–12 days (Fig. 8a). Tumor weight was reduced from a mean of 580 to 370 mg in mice with Rasal1 KD T cells relative to scrambled RNAi. Furthermore, there was a fourfold increase in percent of TILs expressing CD8 + in the tumors of mice injected with Rasal1 KD T cells relative to scrambled RNAi (i.e., 8 vs 38% of CD8 + TILs) (Fig. 8b). By contrast, no change in the percent of CD8 + T-cells was observed in the spleens of mice (right panel). Furthermore, we observed an increase in the percentage of CD8+ T-cells expressing the activation markers CD25 and CD69 in the TIL population in mice injected with Rasal1 KD relative to scrambled RNAi expressing T cells (Fig. 8c). By contrast, the expression of CD62L or CD44 in TILs was unaffected by Rasal1 KD (Fig. 8d). Similarly, an increase in the percentage of CD8+ TILs expressing GZMB IL-2, and IFNγ1 was seen (Supplementary Fig. 3). Overall, Rasal1 KD promoted the expansion of CD8 + TILs in EL-4 tumors with an increased expression of activation markers, accompanied by the reduced growth of solid tumors in mice.

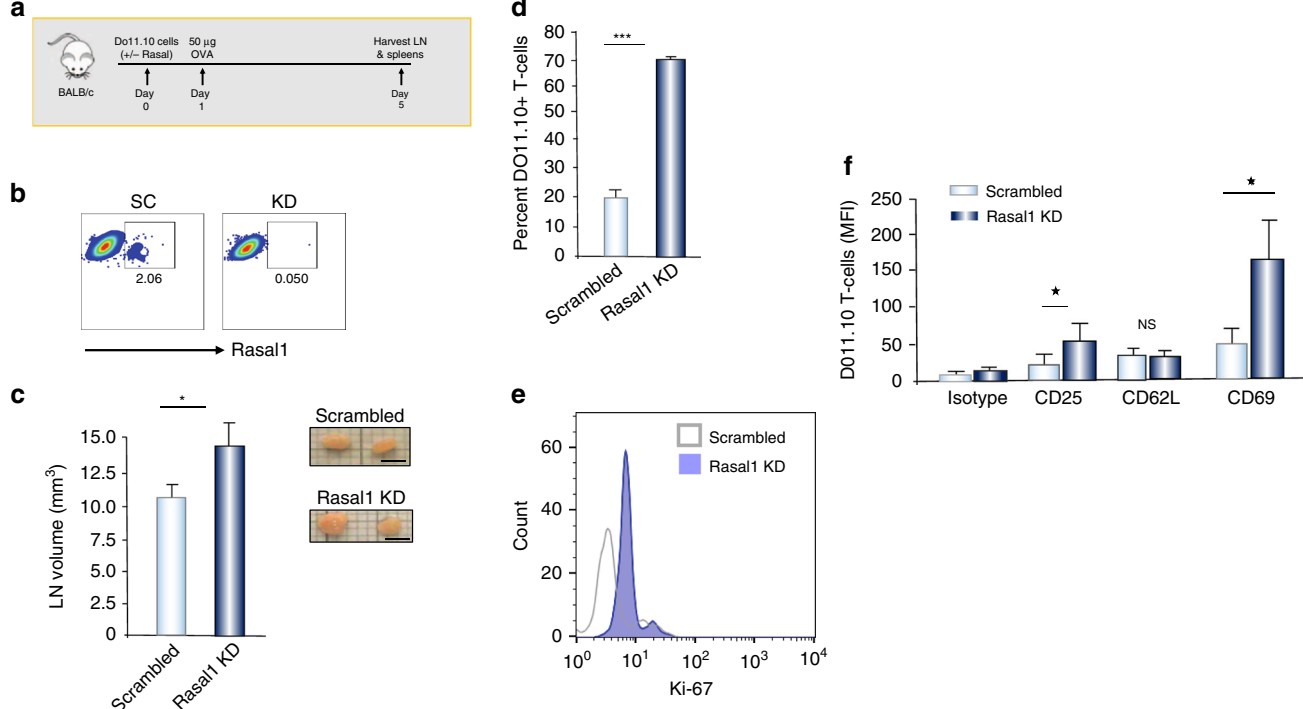

**Fig. 6** Rasal1 inhibits in vivo T-cell activation by peptide antigen. **a** Upper: schematic of DO11.10 T-cell injection followed by OVA peptide injection and an assessment of LNs at day 5. **b** FACS profile of Rasal1 expression in T cells transfected with scrambled siRNA (left) and Rasal1 siRNA (right). **c** Rasal1 siRNA KD increases the size of lymph nodes, right inset: examples of LNs (scale bar, 2.5 mm). **d** Rasal1 KD increases the percent of DO11.10 T cells in lymph nodes; **e** Rasal1 KD increases the presence of DO11.10 T cell with the expression of proliferation marker Ki67; **f** Mean fluorescence intensity levels (MFIs) of various receptors. T cells were analyzed by flow cytometry for the expression of CD25, CD62L, and CD69 ($n > 2$)

## Discussion

Present immunotherapy against cancer involves the use of immune checkpoint blockade against PD-1 and other IRs to enhance T-cell responses[2,3]. However, the discovery of new receptors and intracellular signaling pathways is needed for more effective cancer immunotherapy. Here, we identify a novel pathway involving Rasal1, which associates with the T-cell receptor following T-cell activation and which negatively regulates the p21$^{ras}$-ERK pathway in T cells. Rasal1 negatively regulated CD4 + T-cell responses to antigenic peptide and CD8 + responses against the growth of B16 and EL-4 tumors, the latter event accompanied by the increased infiltration and functionality of CD8 + TILs. Our findings identify a novel TCR-associated negative regulator of T-cell activation that has therapeutic potential in cancer immunotherapy.

Peptides corresponding to Rasal1 were detected using tandem affinity purification TCR zeta tandem affinity purification in conjunction with mass spectrometry. This two-step TAP approach minimizes nonspecific contaminating material, and has been used successfully to purify other complexes with a high level of purity[48,49]. To our knowledge, this is the first study to use the TAP assay to identify new proteins associated with the TCR complex. Aside from the expected TCR and CD3 subunits, Rasal1 was detected in multiple experiments ($n = 3$). This finding underscores the effectiveness of the TAP approach in the identification of protein–protein interactions. Rasal1 was found to be expressed weakly in resting T cells, and instead, is induced as a consequence of the activation of CD4 and CD8 + T cells. It may, therefore, play its main inhibitory role after the initial activation of T cells. This inducible expression is reminiscent of IRs such as CTLA-4 and PD-1 which are also expressed as a consequence of T-cell activation[2].

We further showed that Rasal1 is unusual in having the potential to inhibit T-cell activation via at least two mechanisms. First, we showed that Rasal1 binds to the kinase domain of ZAP-70, and not other key regions of ZAP-70 such as the SH2 domains or the regulatory SH2 intervening region. The binding to ZAP-70 explains why Rasal1 is co-precipitated with the TCR since ZAP-70 binds to the CD3 and zeta chains of the complex[7,9]. PLA analysis clearly showed that the interface of Rasal1 and ZAP-70 depends on ligation of the TCR complex. However, we cannot exclude that Rasal1 may also bind to free ZAP-70, independent of its binding to the TCR complex. Importantly, Rasal1 binding to the ZAP-70 kinase domain inhibited its kinase activity as shown by the reduced autophosphorylation and the phosphorylation of the downstream substrate LAT. To our knowledge, this study is the first reported identification of a GAP protein that binds the ZAP-70 kinase domain to inhibit its catalytic activity. In this context, it is interesting that some mutations in the catalytic domain of ZAP-70 have been implicated in a blockade of thymic differentiation and in autoimmunity[10,60]. Whether the level of Rasal1 inhibition of ZAP-70 is sufficient to elicit effects on thymic differentiation is unclear; however, it may help protect against excess inflammation or autoimmunity and influence T-cell responses to antigens of varying affinities. In this vein, we noted that Rasal1 preferentially inhibits responses to higher concentrations of anti-CD3. Rasal1 may act, therefore, to control overstimulation and inflammation due to high strength TCR signals[61].

The second mode of inhibition involved the intrinsic GAP activity of Rasal1 and its inhibition of the p21$^{ras}$-ERK pathway. We found that the transfection and expression of Rasal1 abrogated the phospho-activation of Raf1 (MEK3K), MEK1, and ERK, but not p38, in response to TCR ligation. There are, therefore, two

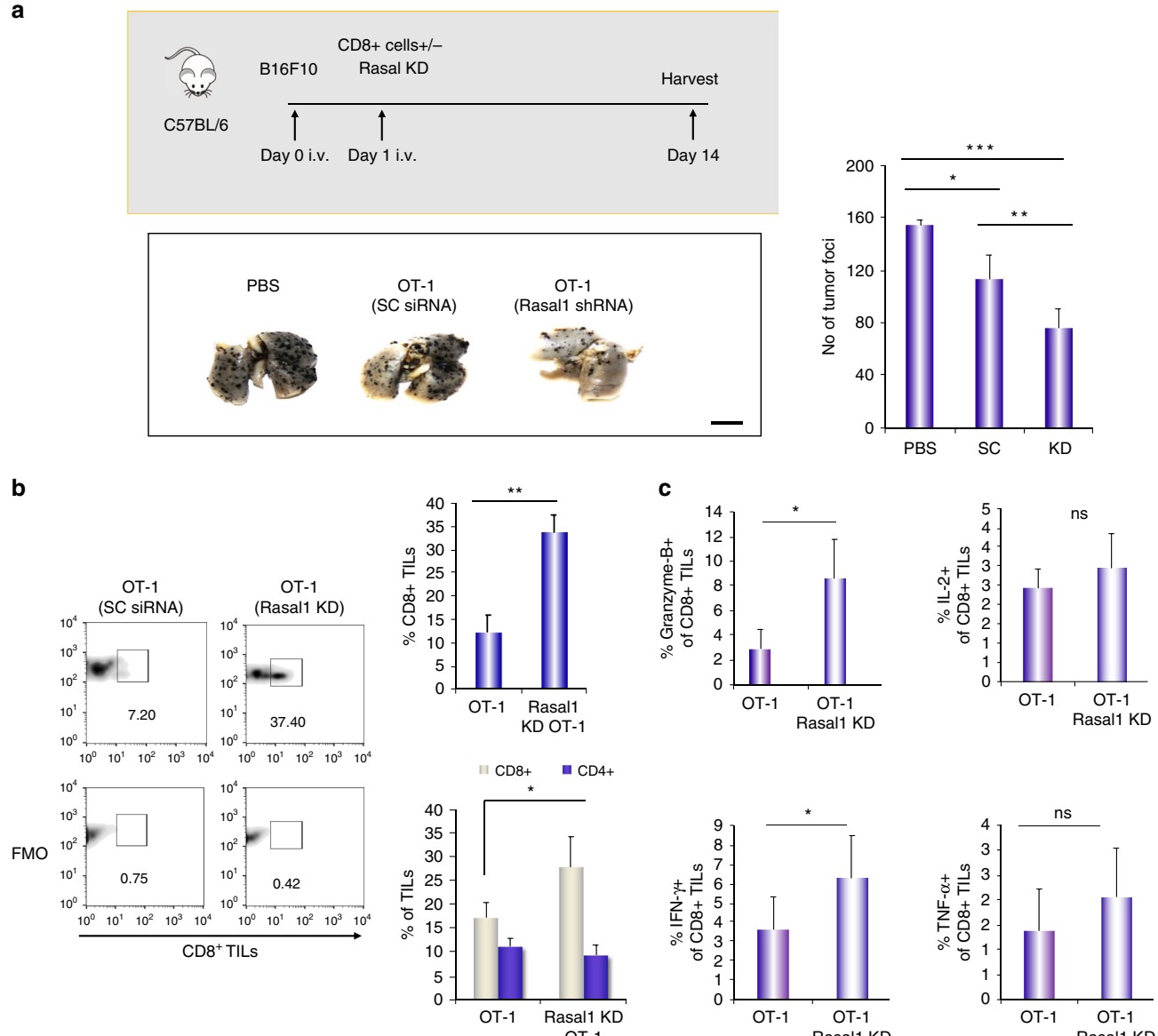

**Fig. 7** Rasal1 KD T cells show enhanced rejection of B16 tumors and the increased presence of activated CD8 + TILs. **a** Schematic of B16 melanomas tumor experiment. Histogram of B16 tumor foci at day 14 in control versus or mice receiving OT-1 cells with control scrambled esiRNA or with Rasal1 esiRNA ($n = $ 4 per group, upper panel). Lower panel: representative image of B16 tumor lungs from each group. Right panel: histogram showing Rasal1 KD reduction of tumor foci in pulmonary B16 metastasis (numbers of tumor foci in lungs) ($n = 3$) (scale bar, 10 mm). **b** Rasal1 KD adoptive cell transfer results in a major increase in CD8 + TILs in B16 tumors. Left panel: FACs histograms showing presence of CD8 + TILs in tumors. SC  scrambled siRNA, Rasal1 KD Rasal1 siRNA. Right upper panel: histogram showing percentage of CD8+ TILS; lower right panel: percentage (%) of CD4 vs CD8 T-cells in lung tumors of B16 animals. **c** Rasal1 KD ACT results in an increase in the GZMB and interferon gamma expressing CD8 + TILs in B16 tumors. Histograms showing % of granzyme B (GZMB), IFN-γ, IL-2, and TNF-α expressing CD8 + TILs. Plots show mean with ± SD. *p*-values were calculated by unpaired *t* test. *$p < 0.05$, **$p < 0.01$, ***$p < 0.001$

pathways by which Rasal1 negatively influences T-cell activation underscoring its potential importance. The degree to which the inhibition of ZAP-70 versus p21ras regulate T-cell activation and effector functions remain for future studies. Rasal1 is a particularly exciting TCR-associated upstream candidate for the anergy induction in transplantation and autoimmunity, given its defined role of the p21ras-ERK pathway in T-cell unresponsiveness[32]. It also should be noted that despite our focus on p21ras, Rasal1 has also been reported to have GAP activity toward the Ras-related small GTPase, Rap[40]. Ras and Rap have somewhat different functions. While Ras regulates proliferation, transcription, and cell survival, Rap has been mostly implicated in adhesion and

cell–cell interactions. We previously showed that an adaptor SKAP1 couples the TCR to the formation of a complex between Rap1 and its ligand RapL for T-cell adhesion[17,18]. It is an intriguing possibility that Rasal1 binding to ZAP-70 of the TCR might also negatively influence Rap1 activity in its control of T-cell adhesion and motility. In this manner, the GAP protein may have more than two roles in the regulation of T-cell immunology.

The GAP activity of Rasal1 differs from the mode of action of other established negative regulators, such as the SHP-1/2 phosphatases and the E3 ubiquitin ligases. SHP-1/2 dephosphorylate substrates[62], and E3 ligases such as Cbl-b, Itch and Grail regulate the ubiquitin pathway[25]. Perhaps the most similar mediator is the

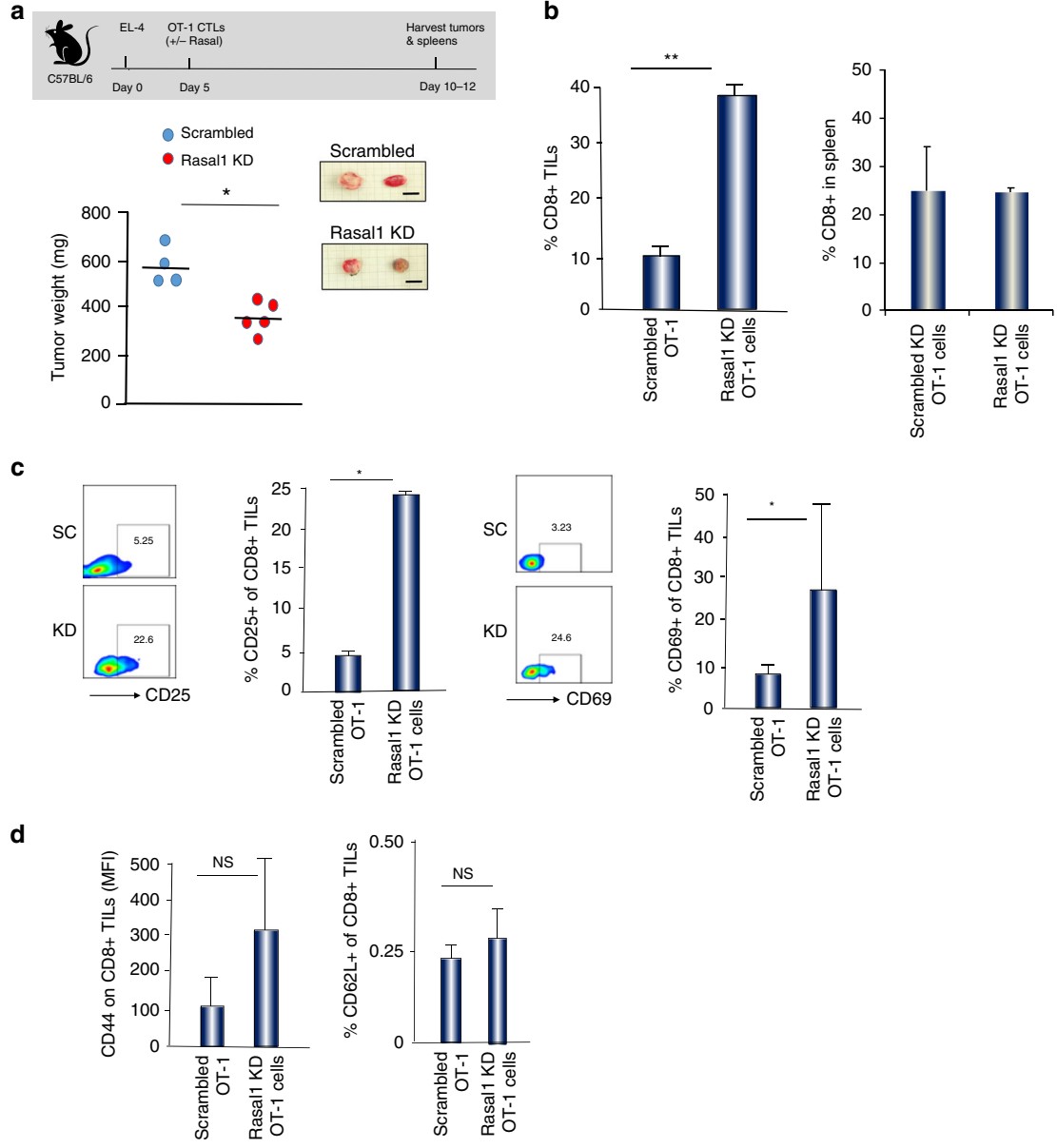

**Fig. 8** Rasal1 KD T cells show reduced EL-4 tumor size concurrent with an increase in the presence of activated CD8 + TILs. **a** Upper panel: schematic of EL4 solid tumor experiment. Lower panel: quantification of tumor size in mice receiving OT-1 cells expressing control scrambled siRNA or Rasal1 KD siRNA T cells ($n = 4$ per group, lower panel). Upper inset panel shows representative images of some EL4 tumors from each group (scale bar, 5 mm). **b** Histogram showing an increase in the percentage of CD8+ TILs in mice injected with CD8 + Rasal1 KD relative to scrambled siRNA T cells (left panel). No increase was seen in the spleens of these mice (right panel). **c** Histograms showing an increase in the expression of CD25 and CD69 on CD8 + TILs from mice injected with control scrambled siRNA or with Rasal1 KD siRNA T cells. **d** Histograms showing the MFI values for CD44 expression (left panel) and the percentage of CD62L-positive CD8 TILs (right panel) from mice injected with scrambled siRNA or Rasal1 KD siRNA T cells

CAPRI ($Ca^{2+}$ promoted Ras inactivator), which is also a $Ca^{2+}$-dependent GAP that switches off the Ras-MAPK pathway following stimuli that elevate intracellular $Ca^{2+}$ [40]. Another less direct pathway involves Cish, a suppressor of cytokine signaling (SOCS), which inhibits PLCγ1 and Ras-ERK signaling[63]. Cish is part of a multimolecular E3 ubiquitin ligase complex with an SH2 domain is essential for PLCγ1 regulation in TCR-stimulated $CD8^+$ T cells[64]. While Cish is primarily cytoplasmic and does not cluster at the plasma membrane upon stimulation, Rasal1 has C2 domains to bind phospholipids in membranes, and which we showed, translocates to membranes upon TCR ligation.

Lastly, we showed that the adoptive cell transfer of T cells expressing Rasal1 siRNA limited the growth of two different tumor types in mice. B16 tumor cells were injected intravenously

into mice and pulmonary metastasis assessed, while EL-4 cells were injected subcutaneously and monitored for solid tumor growth. The reduction in tumor growth was accompanied by a marked increase in CD8 + TILs. In the case of B16 tumors, there was a threefold increase in the presence of OT-1 CD8 + T cells in tumors from mice injected with Rasal1 KD T cells. Further, these cells showed an increase in the expression of GZMB and IFNγ1, the effectors of CD8 + CTL killing of tumor targets[65]. In the case of EL4 tumors, there was also a significant increase in the numbers of OT-1 Rasal1 KD TILs relative to OT-1 scrambled KD TILs, as well as an increase in TILs expressing CD25 and CD69. In both tumor models, the level of CD62L expression was unaffected suggesting that the TCR-Rasal1 pathway does not play a role in the balance between memory vs effector T cells. Instead, it

may promote T cell entry into tumors and help arm CD8 + T cells for more effective tumor killing. We reported similar effects mediated by the GSK-3 pathway in T cells[57,58]. Overall, our findings identify a novel TCR-associated protein that negatively regulates T-cell activation and antitumor immunity. Rasal1 inhibition could improve adoptive immunotherapies using tumor antigen specific T cells[66]. siRNAs, CRISPR-Cas9, or SMIs of the Rasal1 pathway could be used in ACT of T cells that express tumor-specific T-cell receptors (TCRs) or chimeric antigen receptors (CARs). Rasal1 and its binding to ZAP-70 and regulation of the p21$^{ras}$ pathway may serve as a target for the development of novel immunoregulatory drugs.

## Methods

**Cells, reagents, and antibodies**. The 3A9 mouse T-cell hybridoma was obtained from ATCC CRL-3293 (Manassas, Virginia, USA). AP-MS buffers, cell lysis buffer (20 mM Tris-HCl, pH 7.4, 1 × protease inhibitor mixture (Roche Applied Science), 50 mM NaCl, 1% (v/v) NP-40, 1% n-dodecyl β-D-maltoside (Calbiochem) or 1% digitonin (Calbiochem), 1 mM sodium fluoride, 1 mM PMSF (Sigma), 1 mM beta-glycerophosphate (Calbiochem)); wash buffer 1 (10 mM Tris, pH 7.4, 50 mM NaCl, 0.1% (v/v) NP-40, 0.1% n-dodecyl β-D-maltoside, 0.1 mM imidazole (Sigma)); wash buffer 2 (10 mM Tris, pH 7.4, 50 mM NaCl, 0.1% (v/v) NP-40, 0.1% n-dodecyl β-D-maltoside); elution buffer 1 (10 mm Tris, pH 7.4, 50 mM NaCl, 0.1% (v/v) NP-40, 0.1% n-dodecyl β-D-maltoside 1 × protease inhibitor mixture, 0.1 mg/ml of PMSF, 1–10 mM Imidazole (Sigma) gradient); elution buffer 2 (10 mm Tris, pH 7.4, 50 mm NaCl, 0.1% (v/v) NP-40, 0.1% n-dodecyl β-D-maltoside 1 × protease inhibitor mixture, 0.1 mg/ml of PMSF, 2.5 mM D-desthiobiotin (IBA technology)). Rasal1 cDNA clones (Source Bioscience, Cambridge) were amplified and inserted into Srα-vector with Flag- or HA-tag at the N-terminus of protein. pNFAT3-Luc plasmid (contains three tandem repeats of the NFAT/AP1-binding site at ~1287 bp of the murine IL-2 promoter). Antibodies include anti-ERK1/2 (Cell Signaling, Cat. # 9101 S), anti-zeta chain (Bio-rad Cat.# VMA00220 and BD Pharmingen, Cat. # 554241), anti-Rasal1 (Novus Biologicals Cat # NBP1-32776), anti-ZAP 70 (Cell Signaling Cat#3165), rabbit anti-β-actin (Sigma, Cat.#A2228), 2C11 (BioxCell Cat. #BE0001-1), OKT3 (BioxCell Cat.#BE0001-2), p38 (Cell Signaling Cat.#9212), anti-Flag tag (Sigma Cat.# F1804), and anti-HA tag (ThermoFischer Cat.#26183). All antibodies were used at 1:1000, except anti-β-actin (1:2000) for western blotting. For flow cytometry antibodies were used at 1:100 dilutions.

**Mice and T-cell enrichment**. C57BL/6, DO11.10, BALB/c, OT-I, and Rag$^{2-/-}$ mice were housed at the Central Biological Services (Cambridge University). Animal studies were conducted in a gender and age-matched manner for all experiments. Both male and female mice were used, and were 6–8 weeks of age at the time of experiments. T cells were enriched from splenocytes, and LNs using a negative selection column kit (R&D Systems) or MACS microbeads magnetic beads (Miltenyi Biotech). Purity of isolated T cells was >90%.

**TAP for MS**. TCR-null zeta$^{-/-}$ 3A9 cells were reconstituted with HIS-STREP-II-tagged Zeta to reintroduce surface TCR expression (Supplementary Fig. 1). Stable clones were assessed for optimal surface and intracellular TCR abundance in comparison with WT–control cells. A two-step sequential chromatography approach was used to capture TCR and associated protein upon CD3 stimulation under native non-denaturing conditions. Briefly, clarified lysates were loaded on Pro-Bond Ni$^{2+}$-NTA column (Invitrogen) followed by a second purification step, Strep-Tactin sepharose column (IBA Lifesciences). Strep-II-tag protein fractions were gravity eluted with D-desthiobiotin (IBA Lifesciences). Purity was assessed by SDS gel and silver staining. TCR complex was enriched by concentration on a 100 kDa cutoff membrane (Amicon). For MS, gel fragments were destained in ddH$_2$O and 20 mM NH$_4$HCO$_3$, reduced with 2 mM DTT, and alkylated with 10 mM iodoacetamide prior to overnight digestion with 2 μg of sequencing grade trypsin (Promega). ESI–MS was carried out with a 6130 Quadrupole spectrometer, and MS acquisition carried out in positive ion mode, total protein masses were calculated by deconvolution within the MS Chemstation software (Agilent Technologies). The raw MS data files were searched against the UniprotKB Mus musculus database using the Mascot search algorithm (version 2.2.07, Matrix Science, London, UK). Significance threshold was set to $p > 0.05$ and ion score cutoff was set to 0. Protein abundance was estimated by calculating emPAI scores. Proteins identified by at least two unique peptides with an ion score of >30 were considered to be present, which represented a 5% random match probability.

**Flow cytometry, ex vivo stimulation, RT-PCR, and cellular fractionation**. For flow cytometry, surface and intracellular staining cells were processed as described previously[67]. For ex vivo stimulation, TILs or splenocytes were stimulated with PMA/Ionomycin for 6 h at 37° with Golgi Plug (BD). Real-time PCR was performed using commercially available Taqman probe sets (Invitrogen) on ABI PRISM 7000 SDS (Applied Biosystems, USA) instrument as per the manufacturer's

instructions. RNA levels were normalized to endogenous housekeeping 18s rRNA levels and plotted as a fold increase from resting cells. The results were analyzed using SDS 1.2 software (Applied Biosystems, USA). Cytosolic and membrane fractions were isolated from primary T cells, as described previously[18].

**Transfections, antigen presentation, transcription, and proliferation assays**. Primary T cells were transfected by microporation using Amaxa Nucleofector Kit (Lonza) as per the instructions supplied with modification: cells were kept on ice for 10 mins ice and in Amaxa solution A plus B (4.5:1) plus plasmid(s) for 1 min prior to transfection. Antigen presentation and live cell imaging were done using naive DO11.10T cells cultured at a density of $2 \times 10^5/200$ μl cells plus mitomycin-C-treated autologous APCs pulsed with ovalbumin (OVA$_{323-39}$) peptide. In vitro proliferation was also assessed by directly pulsing with 1 μCi of [3 H] thymidine during last 8 h of incubation. For transcription assay, luciferase reporter containing NFAT-AP1 binding sites was co-transfected with control Renilla plasmid pRL-TK (Promega) or Flag-Rasal1 for 24 h. Cell were anti-CD3 stimulated for additional 6 h and activity measured using dual luciferase assay kit (Promega) on a MicroLumat Luminator (Berthold).

**Endogenous Rasal1 KD**. We used MISSION esiRNA (Sigma, Cat. No. EMU036111-50UG) that are endo-ribonuclease prepared siRNA. MISSION esiRNA are a heterogeneous mixture of siRNAs that all target the same mRNA sequence, in part based on previous shRNA knockdown studies[68]. These multiple silencing triggers lead to highly specific and effective gene silencing. esiRNA cDNA Rasal1 target sequence is given below:

GACGTGTCTGGAAGCAGTGACCCCTATTGTCTGGTGAAAGTGGATGACCAAGTGGTGGCCAGGACAGCAACCATCTGGAGGAGCCTGAGCCCCTTTTGGGGGGAGGAGTACACCGTTCACCTTCCATTGGACTTCCACCACCTGGCCTTCTACGTGCTGGATGAGGACACCGTTGGACACGATGACATCATTGGGAAGATCTCATTGAGCAAAGAGGCGATCACAGCCGACCCTCGAGGGATCGACAGCTGGATCAACCTGAGCCGAGTGGATCCAGACGCTGAAGTACAGGGTGAGGTCTGCCTGGATGTGAAGCTATTGGAGGATGCTCGGGGCCGCTGCCTCCGCTGCCACGTGAGACAGGCCAGGGACCTGGCCCCCCGGGACATCTCTGGCACATCGGACCCATTTGCCCGTGTGTTCTGGGGCAACCATAGTTTGGAAACTTCGACCATCAAGAAGACCCGCTTTCCACACTGGGATGAGGTGTTG.

**esiRNA scrambled cDNA target sequence**. GTGAGCAAGGGCGAGGAGCTGTTCACCGGGGTGGTGCCCATCCTGGTCGAGCTGGACGGCGACGTAAACGGCCACAAGTTCAGCGTGTCCGGCGAGGGCGAGGGCGATGCCACCTACGGCAAGCTGACCCTGAAGTTCATCTGCACCACCGGCAAGCTGCCCGTGCCCTGGCCCACCCTCGTGACCACCCTGACCTACGGCGTGCAGTGCTTCAGCCGCTACCCCGACCACATGAAGCAGCACGACTTCTTCAAGTCCGCCATGCCCGAAGGCTACGTCCAGGAGCGCACCATCTTCTTCAAGGACGACGGCAACTACAAGACCCGCGCCGAGGTGAAGTTCGAGGGCGACACCCTGGTGAACCGCATCGAGCTGAAGGGCATCGACTTCAAGGAGGACGGCAACATCCTGGGGCACAAGCTGGAGTACAACTACAACAGCCACAACGTCTATATCATGGCCGACAAGCAGAAGAACGGCATCAAGGTGAACTTCAAGATCCGCCACAACATCGAGGACGGCAGCGTGCAGCTCGCCGACCACTACCAGCAGAACACCCCCATCGGCGACGGCCCCGTGCTGCTGCCCGACAACCACTACCTGAGCACCCAGTCCGCCCTGAGCAAAGACCCCAACGAGAAGCGCGATCACATGGTCCTGCTGGAGTTCGTGACCGCCGCCGGGATCACTCTCGGCATGGACGAGCTGTA.

**Phospho-ERK and p38 assay**. Primary T cells transfected with control or Flag-tagged Rasal1 were stimulated with anti-CD3 antibodies 2C11 (2–5 μg/ml) plus cross-linking RαM (1.25 μg/ml). Reaction was stopped with cold Stop Solution (PBS containing 1 mM Sodium Vanadate plus 1 mM EDTA). PMA (50 ng) and ionomycin (1 μg/ml) was used as control. Cells were fixed and permeabilized with methanol, and phospho-staining performed.

**In situ proximity ligation assay**. Proximity ligation assay was performed using Duolink in situ PLA reagents[50] on Jurkat cells. Cells on slides were blocked with Duolink Blocking stock followed by the application of two PLA probes in 1 × antibody diluent. The slides were washed in a wash buffer (1 × TBS-T) for 5 min twice and processed for hybridization using Duolink Hybridization stock 1:5 in high purity water and followed by incubation for 15 min at 37 ℃. Duolink Ligation was performed with ligase, and the slides were incubated in a pre-heated humidity chamber for 15 min at 37 ℃. Amplification was then achieved using Duolink Amplification stock containing the polymerase, and the slides were incubated again in a pre-heated humidity chamber for 90 min at 37 ℃. DNA was stained with DAPI. One individual dot represents the close proximity of two interacting proteins within the cells.

**Tumor experiments**. The research was regulated under the Animals (Scientific Procedures) Act 1986 Amendment Regulations 2012 following ethical review by the University of Cambridge Animal Welfare and Ethical Review Body (AWERB) Home Office UK PPL No. 70/7544. We complied with all ethical regulations for the

work with mice. For lung tumors, B16F10 melanoma cells were pulsed with OVA$_{257-264}$ overnight, trypsinized, and resuspended in PBS, and then $2 \times 10^5$ cells (in 0.2 mL) were injected via the lateral tail vein on day 0. Mice were injected with OT-I cells or PBS alone on day 1. Mice were killed day 14, and tissues were isolated. Some lungs were also immersed in Bouin solution to distinguish white tumor colonies from yellowish lung parenchyma. Surface metastatic foci in lung left lobes were counted manually. For establishing EL-4 lymphoma tumors, at day 0, $10^5$ EL-4 cells were injected intradermally into the right flanks of mice. Five days after tumor implantation, control esiRNA OT-I or Rasal1 esiRNA transfected OT-I cells were given intravenously. Four mice per group bearing EL4 tumors were kiled when tumor from control mouse (OT-I cells alone) reached limit (12 mm). Tumors were sampled for CD8 TILs infiltration and activation markers.

**Western blotting**. Cells were transfected and 24 h lysed with RIPA buffer (50 mM Tris-HCL pH 8,0, 150 mM NaCl, 1% Triton X-100, 0,5% Na-desoxycholate, 01.% SDS, 1 mM Na$_3$VO$_4$, 1 mM PMSF, 1 mM DTT, and protease inhibitor complete (Roche). Polyvinylidene difluoride (PVDF) membranes were used for western blotting applications. Blocking of membranes was performed with TBST, including 2% BSA. The membrane was incubated with the first antibody (1:1000) for 1 h or overnight followed by HRP-conjugated secondary antibodies (1:5000) (GE Healthcare). The ECL western blotting substrate was used for detection.

**In vitro kinase assays**. In vitro kinase assays were performed using 10× PK buffer (New England Biolabs) supplemented with 0.05 mM ATP and 1 µCi of (γ-$^{32}$)ATP (3000 Ci/mmol; Amersham Pharmacia) for 30 min at 37 °C in the presence of recombinant human active ZAP-70 Kinase (ProQuinase), as previously described[6,7]. Bacterially expressed purified GST proteins or Flag-expressed proteins from 293T cells served as substrates. Flag-LAT used in the kinase assays was immunoprecipitated from 293T cells. Samples were resolved by SDS/PAGE and subjected to autoradiography or western blotting.

**Immunoprecipitation**. Cell lysates were incubated with Protein-G sepharose beads (GE Healthcare) and the specific antibody overnight at 4 °C. Immunoprecipitates were washed three times with ice-cold buffer (20 mM Tris-HCl ph 8.2, 150 mM NaCl, 1% (v/v) TritonX-100 and analyzed by western blotting[17,69,70].

**Statistical analysis**. All statistical analysis was performed using Prism 6 software (GraphPad Software). An ANOVA test was used to test the significance of the changes between groups.

## Data availability
The authors declare that the data supporting the findings of this study are available within the article and its supplementary information files. Any further information on the proteomics data is available from author on request.

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

## Acknowledgements

C.E.R. was supported by the Canadian Institutes of Health Foundation grant (159912), while R.T, M.R. and C.E.R. were supported by the Wellcome Trust (092627/Z/10/Z). M.R. and K.S. received support from the Deutsche Forschungsgemeinschaft, the Research Support Foundation, the Messer Stiftung, the Deutsche Krebshilfe and the BANSS Stiftung, and the German Cancer Consortium (DKTK), (Heidelberg). We thank Drs. Mark Issa, Hien Thai Tu (Centre de Recherche, Maisonneuve-Rosemont Hospital, Montreal, Quebec H1T 2M4, Canada) and Silvia Guil Luna, Centre de Recherche, Maisonneuve-Rosemont Hospital, Montreal, Quebec H1T 2M4, Canada; Instituto Mai-mónides de Investigación Biomédica de Córdoba, Spain) for helpful suggestions in reading and editing the paper).

## Author contributions

R.T., M.R., and C.E.R. designed different aspects of the research. R.T. and M.R. conducted the majority of experiments, while R.T., M.R., K.S. and C.E.R. drafted the paper.

## Competing interests

The authors declare no competing interests.
