## [Peer Review File · Nature Communications]

Reviewers' comments:

Reviewer #1: T cell signalling
(Remarks to the Author):

RAS protein activator like 1 (RASAL1) is a novel TCR associated mediator than (sic) negatively regulates T-cell proliferation and antitumor mediated immunity

Thaker et al.

Manuscript ID: NCOMMS-18-13325

In this manuscript the authors describe a potential role for GTPase activating protein RASAL1 in downregulation of T cell signaling. They show that it binds to the T cell receptor and/or the protein tyrosine kinase ZAP-70 and that it downregulates ERK activation. Finally, they show that knockdown of Rasal1 expression in an adoptive transfer model increases tumor clearance.

There are some interesting results in the study, but many experiments are missing and some are contradictory. Following are a series of questions, figure by figure:

1. Figure 1. Please give a better description of 3A9 cells. Are they T cells? Later data beg the question of whether 3A9 cells bear ZAP-70. I assume that these cells are not activated, but that's not described and later data again beg that question. The list of peptides in Fig. 1A is grossly deficient. What other peptides were identified? Is a ZAP-70 peptide present? What other TCR signaling proteins were identified or not identified? These are central questions about the specificity of the experiment and results. Why is a list of the entire MS data not show? Figure 1C – the authors need to include controls for their membrane and cytosolic fractions.

2. In Fig 2 the authors imply that the reader really should not care what was presented in Fig. 1. It's actually ZAP-70 that's associated with Rasal1. Does Rasal1 even co-immunoprecipitate with TCRzeta? It's not shown. Note in the first paragraph the protein is described as Rasal1; in the second paragraph it's RASAL1 and in the third it's Rasal1.

In Fig. 2A, all panels except for the bottom right do not appear to have DAPI-stained cells in the field. Authors need to show that there are cells in the field, just not positive for the interaction. Also, several of the interacting RASAL-ZAP70 spots appear to be cytosolic. Authors should comment on this. Fig. 2C – they should show a western blot for Rasal1 to show that the IPs worked, and similar levels of the protein were IP'd.

3. Figure 3 contains overexpression data indicating that Rasal1 can decrease Erk activation, but not data that at physiologic levels it does. Authors should quantify the pERK divided by total ERK in the westerns. Also it is unclear why they have included the Flag-Plk1 expressing cells in the figure – they do not mention this at all in the text. This figure is poorly described.

4. There's more overexpression data and finally some knock-down data for Rasal1. What's missing is any detailed biochemical analysis in the overexpression and knock-down studies. Are activated Ras levels affected? What is the effect on kinases between Ras and Erk, i.e. Raf and Mek? Figure 4D: the western blot upper panel is labeled anti-Flag-Rasal1 – but the authors are knocking down endogenous Rasal1 with siRNA. They need to explain what they are detecting here.

5. Figure 6C: it is unclear how the authors analyzed the expression of activation markers. Did they stain for the various markers and gate on the CD8+ TILs? Also what is FMO?

Reviewer #2: T cell signalling and cancer

(Remarks to the Author):

In this manuscript, Thaker et al identify a novel interaction between Rasal1 and components of the TCR, initially by co-precipitation with the zeta chain but found to be more intimately associated with ZAP70. Rasal1 is part of a larger family of Ras GAP activating proteins. No function for Rasal1 in T cells has been described so far in the literature.

Figure 2 shows a strong PLA signal between RASAL and ZAP70 and this is confirmed to some extent by immunoprecipitations (the band in Fig 2C is very weak and no controls are included). These assays are all non-quantitative and it is unclear how strong the interaction between ZAP70 and Rasal1 compares to other ZAP70 interaction partners. No effort has been made to describe the interaction between Rasal1 and ZAP70 (eg which domains of each protein is involved. This should be addressed using recombinant proteins.

As might be expected, overexpression of Rasal1 led to reduced pErk. The use of a catalytic dead mutant would be a good control to include in this experiment. The experiment should also be complemented by knocking out Rasal1.

Functionally, overexpression of Rasal1 led to reduced T cell proliferation whereas (incomplete) knockdown using siRNA increased proliferation about 2 fold. A CRISPR/Cas9 approach might have provided more definitive results. Moreover, the siRNA oligo sequences used for the knockdowns should be included.

siRNA knockdown of Rasal1 enable better control of tumours (B16 and EL4, both expressing Ova, presumably by increasing expression of IFN γ and GzmB. These are promising results which support the key role for Rasal1 as a negative regulator of the Ras pathway in T cells.

The authors describe the inhibition of negative regulators of a signaling protein as a novel approach. A similar concept has been shown for Cish which inhibits PLC γ and hence also impacts on Ras-Erk signalling (Palmer et al, JEM 2015). The data presented in the current manuscript highlights the previously unknown role of Rasl1 and shows that Ras-Erk is most likely to be the most relevant target. However, the authors should cite the previous work by Palmer et al. and present their current work in light of their findings.

There is currently no available Rasal1 knockout mouse; however, with the advent of CRISPR/Cas9 it can be argued that the use of siRNA in the present study is not the most rigorous way to test for loss of function of Rasal1.

Reviewer #3: Cancer immunology
(Remarks to the Author):

In this manuscript by Thaker et al., the authors found that Rasal1 is upregulated upon TCR stimulation by anti-CD3 antibody and it associates with TCR complex via ZAP-70. Overexpression of Rasal1 inhibits phosphorylation of ERK upon anti-CD3 stimulation in primary mouse T cells. In vitro proliferation of DO11.10 CD4 $^+$ T cells with OVA peptide or T cells stimulated with anti-CD3 antibody was inhibited in cells overexpressing Flag-Rasal1. Upon knockdown of Flag-Rasal1, these cells showed increased proliferation upon anti-CD3 antibody stimulation. Rasal1 knockdown DO11.10 T cells, when transferred into BALB/c mouse showed increased percentage of these cells after OVA stimulation. In tumor models, Rasal1 knockdown OT-1 CD8 $^+$ T cell showed higher frequency of CD8 $^+$ T cells and cytokine producing cells in TILs, and reduced number of B16 melanoma metastasis or the weight of EL4 tumors. These results suggest that Rasal1 may be an important molecule regulating TCR activation. However, there are several major concerns with this study.

The authors used a number of in vitro experiments to demonstrate the effects of Rasal1 overexpression or knockdown; however, how Rasal1 is regulated or functions in a physiological condition is unclear. Given that anti-CD3 antibody mediated activation induces Rasal1 in primary T cells, the authors need to assess its expression level in naïve, effector/memory, or pre-activated cells in CD4+ and CD8+ T cells and determine whether Rasal1 shows similar or different functions in these T cell subsets. Furthermore, effects of Rasal1 modulations should be addressed with polyclonal and conventional T cells in vivo, in addition to the TCR transgenic models.

The manuscript is poorly written especially in the Results section and some parts were incomprehensible. There are also a number of inconsistencies, typographic errors, mislabeled units or figure, and unexplained abbreviations, all diminishing the integrity of the study. It is also required for the authors to indicate the number of experiments performed or replicates for each figure. In the title, I believe "than" should be "that".

Reviewers' comments:

Reviewer #1

“In this manuscript the authors describe a potential role for GTPase activating protein RASAL1 in downregulation of T cell signaling. They show that it binds to the T cell receptor and/or the protein tyrosine kinase ZAP-70 and that it downregulates ERK activation. Finally, they show that knockdown of Rasal1 expression in an adoptive transfer model increases tumor clearance.”
“There are some interesting results in the study”. We thank the Reviewer for the positive comments. Changes have been underlined in the text. Several new figures have been added to address the issues raised by the Reviewers including Fig. 1C, Fig. 3A-F, SFIGs. 1-4.

“Figure 1. Please give a better description of 3A9 cells. Are they T cells? Later data beg the question of whether 3A9 cells bear ZAP-70. I assume that these cells are not activated, but that’s not described and later data again beg that question.” Agreed. 389 are hybridoma T-cells that lack the expression of the zeta chain for the surface expression of TCR. The cells were stably transfected with the zeta chain coupled to tags (Stept-II and 6His/V5) for tandem affinity purification (TAP), as outlined in the Methods. This led to the cell surface expression of the TCR. The zeta chain is needed for the surface expression of the TCR on T-cells (Hall et al., 1991 Int Immunol. 3, 359-68). Clones of 389 T-cells with surface expression were then grown for use in the TAP purification. SFIG. 1 shows the expression of the zeta in one of the clones 1.2 which was used in the purification procedure (middle panel) and the resultant surface expression of the TCR as detected using anti-CD3 staining and flow cytometry (lower panel). The hybridoma cells were activated with anti-CD3 for 5 min prior to their use in TAP. The TAP approach allows for a two-step purification which greatly reduces the presence of non-specific background binding. A description of this and the new SFIG. 1 has been included in the text (Pg 6, 1st para). The gels were then cut into sections for MS/MS analysis. The cutting of the gels occurred in the 70KD region resulting in the omission of ZAP-70 in MS analysis of proteins from the gels. However, as seen in Fig. 1B, precipitation of the TCR complex readily co-precipitated ZAP-70 from these cells.

“The list of peptides in Fig. 1A is grossly deficient. Why is a list of the entire MS data not show?”
We agree and have provided the proteomic datasets from two experiments as an example (new **SFIG. 2**). Aside from the TCR and CD3 subunits, Rasal1 was the only protein to be reproducibly seen in 3 separate purification experiments. The presence of Rasal1 was then validated by qPCR and western blotting.

“Figure 1C – the authors need to include controls for their membrane and cytosolic fractions.”
Agreed. Figure 1C is now Figure 1D and includes the anti-LAT control which is a transmembrane protein that was only found in the membrane fraction. This control demonstrates that the fidelity of the separation of the cytosolic from the membranes.

“2. In Fig 2 the authors imply that the reader really should not care what was presented in Fig. 1. It’s actually ZAP-70 that’s associated with Rasal1. Does Rasal1 even co-immunoprecipitate with TCRzeta? It’s not shown”. We apologise for this omission. As mentioned, the gel for the quantitative analysis with MS/MS was cut into several pieces which excluded ZAP-70 in the 70-80KDa range. However, to show that ZAP-70 associated with TCR from the 3A9 cells, we have now included an immunoprecipitation in which ZAP-70 is visible (**new Figure 1B**). This clearly shows the presence of ZAP-70 in response to anti-CD3 ligation. There is therefore no doubt that ZAP-70 can be co-precipitated with TCR complex from these cells.

We have also added new data on the mapping of Rasal1 binding to ZAP-70 (**Fig. 3**). Rasal1 bound to full-length and the kinase domain but no other domains in ZAP-70 (**Fig. 3B**). Further, we showed that Rasal1 inhibits ZAP-70 kinase activity as monitored by in vitro auto-phosphorylation and the phosphorylation of the ZAP-70 substrate LAT (**Fig. 3C-E**). We also show that the phosphorylation of LAT is reduced in T-cells expressing Rasal1 (**Fig. 3F**). These data provided important new data on the binding site of Rasal1 on ZAP-70 and its inhibitory effect on ZAP-70. To our knowledge, this is first example of ZAP-70 regulating by a mediator that binds to ZAP-70 in T-cells.

“Note in the first paragraph the protein is described as Rasal1; in the second paragraph it’s RASAL1 and in the third it’s Rasal1”. We apologize and now list Rasal1 in lower case throughout the text.

“In Fig. 2A, all panels except for the bottom right do not appear to have DAPI-stained cells in the field. Authors need to show that there are cells in the field, just not positive for the interaction.” . Agreed. All cells within the field now clearly show DAPI staining in Fig. 2A.

“Also, several of the interacting RASAL-ZAP70 spots appear to be cytosolic. Authors should comment on this.” While PLA can show nearest neighbour localisation, it cannot be used accurately to determine the intracellular localization of the complexes. The staining pattern across the cell could be occurring on the surface of the cell overtop of the nucleus and cytoplasm. This point has been added to the text (Pg. 7, 3rd para).

“Fig. 2C – they should show a western blot for Rasal1 to show that the IPs worked, and similar levels of the protein were IP’d.” Agreed. This has been included in Fig. 2C, showing levels of precipitated Rasal1 as well as the input levels of ZAP-70 from which the Ips were conducted.

“Figure 3 contains overexpression data indicating that Rasal1 can decrease Erk activation, but not data that at physiologic levels it does. Authors should quantify the pERK divided by total ERK in the westerns.” Agreed. A histogram showing this quantitation has now be included in new Fig. 4A. The signal was similar to that seen for endogenous Rasal1.

“Also it is unclear why they have included the Flag-Plk1 expressing cells in the figure – they do not mention this at all in the text. This figure is poorly described.” Flag1-Plk1 is a serine-threonine kinase involved in mitosis that was simply provided as a random negative control.

“What is the effect on kinases between Ras and Erk, i.e. Raf and Mek?”. In response to the Reviewer, we have now included blotting of c-Raf on the auto-phosphoylation site Ser338 and on the MEK1 activation site Thr286 (new **Fig. 4C**). Transfected T-cells were ligated for 5 min followed by blotting of cell equivalent lysates with phospho-specific antibodies. While anti-CD3 treatment increased both c-Raf and MEK1 phosphorylation (lanes 2 vs 1, upper and lower panels, respectively). By contrast, CD3 ligation had only a marginal effect in increasing phosphorylation of either substrate (lane 4 vs. 3, upper and lower panels, respectively). These data showed that Rasal1 inhibits CD3 activation of the p21^{ras}-Raf-MEK1-ERK activation pathway in T-cells. This finding combined with the clear sequence homology of Rasal1 with other well-established GAPs such as p120 RasGAP (or Rasa1), NF1, CARPI and Rasa2-4, which regulate p21^{ras}, all supports a connection to p21^{ras} (Liu et al 2005 JCB 170, 183-90; Walker et al 2004 The EMBO J. 23, 1749-60).

We now also show that Rasal1 binds and inhibits the kinase domain of ZAP-70, thereby providing two mechanisms by which Rasal1 can act as a negative regulator of T-cell activation. We showed that Rasal1 inhibits ZAP-70 kinase activity as monitored by in vitro auto-

phosphorylation and the phosphorylation of the ZAP-70 substrate LAT (**Fig. 3C-E**). We also show that the phosphorylation of LAT is reduced in T-cells expressing Rasal1 (**Fig. 3F**). These data provided important new data on the binding site of Rasal1 on ZAP-70 and its inhibitory effect on ZAP-70. To our knowledge, this is a novel mechanism by which ZAP-70 is regulated in T-cells. This description has now been added to the text (Pgs. 8-9).

“Figure 4D: the western blot upper panel is labeled anti-Flag-Rasal1 – but the authors are knocking down endogenous Rasal1 with siRNA. They need to explain what they are detecting here.” Agreed, the Reviewer is correct. This was a typo. It has now been corrected to show “anti-Rasal1”

“Figure 6C: it is unclear how the authors analyzed the expression of activation markers. Did they stain for the various markers and gate on the CD8+ TILs?” Our apology, these expression profiles correspond to percentages of various cell populations gated on CD8+. This has now been made clearer in the new figure.

“Also, what is FMO?” The Fluorescence Minus One Control, or FMO control is a type of control used to properly interpret flow cytometry data. It is used to identify and gate cells in the context of data spread due to the multiple fluorochromes in a given panel. An FMO control contains all the fluorochromes in a panel, except for the one that is being measured.

We have addressed the Reviewers comments and added new data in the form of multiple new figures related to the binding of Rasal1 to ZAP-70, inhibition of kinase activity, expression of Rasal1 in CD4 and CD8+ T-cells, the improved quality of images and precipitations as well as a fuller accounting of the different proteins detected by the tandem sequential precipitation analysis. We hope that the paper is now suitable for publication. It describes a new interaction between the TCR and a negative signaling pathway in T-cells and should be of major interest to members of the T-cell immunology and cancer field.

Reviewer #2:

“No function for Rasal1 in T cells has been described so far in the literature.” “These are promising results which support the key role for Rasal1 as a negative regulator or the Ras pathway in T cells.” We thank the Reviewer for this comment. Changes have been underlined in the text. Several new figures have been added to address the issues raised by the Reviewers including Fig. 1C, Fig. 3A-F, SFigs. 1-4.

“No effort has been made to describe the interaction between Rasal1 and ZAP70 (eg which domains of each protein is involved. This should be addressed using recombinant proteins.” Agreed. In response, we now show that Rasal1 binds and inhibits the kinase domain of ZAP-70, thereby providing two mechanisms by which Rasal1 can act as a negative regulator of T-cell activation. We showed that Rasal1 inhibits ZAP-70 kinase activity as monitored by in vitro auto-phosphorylation and the phosphorylation of the ZAP-70 substrate LAT (**Fig. 3C-E**). We also show that the phosphorylation of LAT is reduced in T-cells expressing Rasal1 (**Fig. 3F**). LAT1 phosphorylation which has been shown to be phosphorylated by ZAP-70 (Zhang et al 1998 Cell 92, 83-92). These data provided important new data on the binding site of Rasal1 on ZAP-70 and its inhibitory effect on ZAP-70. To our knowledge, this is a novel mechanism by which ZAP-70 is regulated in T-cells. This description has now been added to the text (Pgs. 8-9).

“As might be expected, overexpression of Rasal1 led to reduced pErk. The use of a catalytic dead mutant would be a good control to include in this experiment. The experiment should also be complemented by knocking out Rasal1.” It is important to note that we have used a combination of Rasal1 over-expression and siRNA knock-down to document the inhibitory effects of Rasal1 on various parameters such as ERK activation, proliferation in response to anti-CD3 and peptide antigen. Unfortunately, to our knowledge, catalytically dead mutants of Rasal1 have not been identified. We nevertheless feel that the use of siRNA is a more powerful approach since other regions of Rasal1, even catalytically inactive Rasal1, might engage pathways that might skew results. The siRNA KD clearly reduced the expression of Rasal1 allowing an uncomplicated interpretation of the data.

“Functionally, overexpression of Rasal1 led to reduced T cell proliferation whereas (incomplete) knockdown using siRNA increased proliferation about 2-fold. A CRISPR/Cas9 approach might have provided more definitive results.” We agree that other approaches such as CRISPR is possible; however, we also feel that the effects of siRNA in partially reducing Rasal1 expression still had statistically significant effects on various aspects of T-cell function such as in the in vivo expansion of T-cell responses to OVA peptide in the DO11.10 mouse and in response to B16 and EL-4 tumors. The ability to see significant phenotypes with a partial knock-down of a KD is in itself powerful evidence in support of the importance of Rasal1 in T-cells.

“Moreover, the siRNA oligo sequences used for the knockdowns should be included.” Agreed. We used MISSION esiRNA from Sigma which is a heterogeneous mixture of siRNAs that all target the same mRNA sequence common to all transcripts. These multiple silencing triggers lead to highly specific and effective gene silencing. We have added this information to the methods section. We apologize for the omission.

“siRNA knockdown of Rasal1 enable better control of tumours (B16 and EL4, both expressing Ova, presumably by increasing expression of IFN γ and GzmB.” We agree. These are models

based on the extensive evidence from many groups (including our own: Taylor et al 2018 Cancer Research 78(3):706-717) implicating IFN γ and GzmB in CD8+ CTL function.

“The authors describe the inhibition of negative regulators of a signaling protein as a novel approach. A similar concept has been shown for Cish which inhibits PLC γ and hence also impacts on Ras-Erk signalling (Palmer et al, JEM 2015). The data presented in the current manuscript highlights the previously unknown role of Rasl1 and shows that Ras-Erk is most likely to be the most relevant target. However, the authors should cite the previous work by Palmer et al. and present their current work in light of their findings.” We thank the Reviewer for pointing this out and we have now added and commented on this and the CAPRI pathway in the text. We also included statements “The closest analogy is the identification of CAPRI (Ca²⁺ promoted Ras inactivator), a Ca²⁺-dependent Ras GTPase-activating protein (GAP) that switches off the Ras-MAPK pathway following a stimulus that elevates intracellular Ca²⁺ (Lockyer et al 2001). Rasal1 and CAPRI are both calcium sensitive members of the GAP family. Another less direct pathway involves Cish, a suppressor of cytokine signaling (SOCS) family that inhibits PLC- γ 1 which can impact on Ras-Erk signalling (Palmer et al, 2015).” (Pg.16, last para).

We have addressed the Reviewers comments and added new data in the form of multiple new figures related to the binding of Rasal1 to ZAP-70, the inhibition of kinase activity, the expression of Rasal1 in CD4 and CD8+ T-cells, the improved quality of images and precipitations as well as a fuller accounting of the different proteins detected by the tandem sequential precipitation analysis. We hope that the paper is now suitable for publication.

It describes a new interaction between the TCR and a negative signaling pathway in T-cells and should be of major interest to members of the T-cell immunology and cancer field.

Reviewer #3:

“These results suggest that Rasal1 may be an important molecule regulating TCR activation”.

We thank the Reviewer for this comment. Changes have been underlined in the text. Several new figures have been added to address the issues raised by the Reviewers including Fig. 1C, Fig. 3A-F, S Figs. 1-4.

“The authors used a number of in vitro experiments to demonstrate the effects of Rasal1 overexpression or knockdown; however, how Rasal1 is regulated or functions in a physiological condition is unclear.”

We agree that additional data could be added related to how Rasal1 expression is regulated (i.e. different signals, methylation, transcription factors etc.) and additional aspects of its regulation of responses in mice, however, we have already included 3 different in vivo models of how Rasal1 regulates responses (i.e. DO11.10, OT1 B16 and EL-4 models) and this is the first paper with 8 figures already.

We are hoping that this paper would provide an initial description of binding between TCR-ZAP-70 and Rasal1, the sites of interaction and effects on ZAP-70 and the Ras-Raf-MEK1-ERK pathway, as well as the inhibition of responses to anti-CD3, antigen and TIL responses to tumors using 2 tumor models. For future work, we are planning to explore the phenotype of the KIs with mutations that interfere with the interaction as well as the phenotype of the Rasal1^{-/-} mouse which we are presently deriving.

As mentioned, this is the first paper with 8 figures already. We hope that the Reviewer will agree that we have included considerable data for a first paper. If we were to include all functional data with conditional knockouts on TCR specific backgrounds, its potential involvement in T-cell anergy or exhaustion etc. we would presumably be submitting a larger paper to another journal such as Nature Immunology or Immunity. It seems to us that our discovery of a novel inhibitory molecule associated with the TCR is of great potential importance. Rasal1 is a new player in T-cell activation and a potential target in cancer immunotherapy. We hope this is reasonable.

“Given that anti-CD3 antibody mediated activation induces Rasal1 in primary T cells, the authors need to assess its expression level in naïve, effector/memory in CD4+ and CD8+ T cells and determine whether Rasal1 shows similar or different functions in these T cell subsets.”

In response to the Reviewer, we have now included an expression analysis of Rasal1 in T cell subsets, CD4 and CD8 (**new Figure 1C**). There is no detectable Rasal1 expression in naïve CD4 and CD8+ T-cells. The two subsets exhibit differences where Rasal1 expression is more transient in CD4 than CD8 T-cells. This fits with the stronger effect of Rasal1 on CD8+ T-cells in TIL populations. We have included an analysis of CD44 or CD62L expression in response to the EL-4 T-cell lymphoma and B16 melanoma. TILs from EL-4 T-cell lymphoma showed no increase in CD62L (**Fig. 6F**) or in response to B16 tumors which showed no change in the CD44 or CD62L expression (**S Fig. 3**). We see no evidence that Rasal1 is involved in memory differentiation.

“.. effects of Rasal1 modulations should be addressed with polyclonal and conventional T cells in vivo, in addition to the TCR transgenic models.”

We agreed that it would be of interest to assess effects of polyclonal cells; however, this is the first paper to describe the interaction between Rasal1 and ZAP-70 (already 8 figures). The co-precipitation was from polyclonal T-cells (**Fig. 2B**) and we have seen effects on activation using DO11.10 T-cells in response to peptide antigen (**Fig. 4**) and OT-1 T-cells (**Fig. 6**) and use of 2 tumor models. We are hoping that this range of assays in WT and different TCR transgenics combined with the first description of a biochemical interaction between TCR-ZAP-70, its new inhibitory effects on ZAP-

76 activity (**new Fig. 3**) and the ERK pathway would be considered more than adequate for our first paper for Nat Communications

“The manuscript is poorly written especially in the Results section...inconsistencies, typographic errors, mislabeled units or figure, and unexplained abbreviations, all diminishing the integrity of the study.” We have thoroughly proof read the manuscript before.

“It is also required for the authors to indicate the number of experiments performed or replicates for each figure”. This has now been included in the figure legends.

We have addressed the Reviewers comments and added new data in the form of multiple new figures related to the binding of Rasal1 to ZAP-70, expression of Rasal1 in CD4 and CD8+ T-cells, the improved quality of images and precipitations as well as a fuller accounting of the different proteins detected by the tandem sequential precipitation analysis. We hope that the paper is now suitable for publication.

It describes a new interaction between the TCR and a negative signaling pathway in T-cells and should be of major interest to members of the T-cell immunology and cancer field.

Reviewers' comments:

Reviewer #1 (Remarks to the Author):

Thaker et al.

Manuscript ID:

In the revised version of the manuscript the authors have attempted to address my criticisms. However, the signaling experiments are inadequate, the manuscript is still sloppy and there are discrepancies in the results, figure legends and methods as well as missing controls and missing experimental details. Overall at this stage of the submission process to a Nature journal, this seems inappropriate.

For example:

1. The authors call the hybridoma cell 3A9 in the results and 389 in the letter to the referees. Then in Results and legends, they describe that the 3A9 hybridoma clone was activated for 10 minutes. However, in the letter to referees they mention that cells were activated for 5 minutes for the mass-spec experiments.
2. Figure 1C should contain more detailed RNA expression kinetics; three points are insufficient. The figure is still missing a cytosolic fraction control.
3. The authors claim in the letter to referees and in the revised Results that PLA cannot be used to determine location of interacting proteins with no references. However, the description of the Duolink PLA system on the SIGMA website claims that this assay can be used for localization. The authors should substantiate their claims. Additionally, they state that there's no signal when anti-zeta antibodies are used with anti-Rasal1, but there is also no positive control demonstrating that any PLA assay would work with this anti-zeta reagent.
4. In Figures 1 and 2, authors describe that TCR-Rasal1 (Fig. 1) or ZAP70-Rasal1 (Fig. 2) interaction is CD3-dependent. But then in Figure 3 they evaluate ZAP-70 Rasal1 interaction in 293T cells which does not express TCR/CD3. They should comment on this.
5. In the description of Figure 2C in page 8, authors mislabel the lanes.
6. The newly included Figure 3 has many issues. The methods are poorly explained: is the FLAG-LAT used for the assay IP-ed from 293T lysates? There is no evidence that the ZAP fragments are properly folded; do the SH2 domains bind phospho-zeta? They are also missing several control blots. For example: 3B should include Rasal IP blot, 3C should include ZAP-70 and LAT blots, 3E should include a ZAP-70 blot and 3F should include a Rasal1 blot. These are standard controls that should be included.

Further on Fig. 3, standard curves of ZAP-70 phosphorylations should be shown for autophosphorylation and for substrate phosphorylation. Only then can the effects of Rasal1 be understood. As is, the curves in Fig 3C look odd with a dramatic effect in phosphorylation with a 2x increase in Rasal1. Coomassie is misspelled. Fix "GST-Bora had not (sic) effect..." (p.9). The decrease in pLAT in 3F is unconvincing and should be quantified. The mechanism of Rasal1 remains unclear

7. The discussion regarding Erk activation is unclear. What's the hypothesis? That Ras is directly inhibited as Rasal1 is a GAP? That Erk is inhibited because LAT is inhibited? Because ZAP-70 is inhibited? They do some Erk assays and a mediocre Raf and Mek assay. More is needed. Fig. 4C should include total Raf and Mek blots.

Reviewer #3 (Remarks to the Author):

In the revised manuscript by Thaker et al., the authors have adequately addressed the comments raised by the reviewer.

In the description of Figure 2B on page 7 and 8, lane numbers seem different from the data shown in Figure 2B.

Reviewer #4 (Remarks to the Author):

The authors described a new interaction and function of the RASAL1 protein. They established that RASAL1 could interact with ZAP70, more precisely on the kinase domain of this protein. They showed by overexpression or by knockdown that RASAL1 is an inhibitor of ZAP70, capable of regulating T cell proliferation and function in vitro of in vivo. These results are new and interesting. The article is convincing.

Concerning the answers to reviewer 2, the major points raised were answered to, either by new experiments or by additions in the text. The only point left unanswered is the use of CRISPR to knock out the gene more efficiently. However, the comments from the authors make sense, as only a partial knockdown or the protein is sufficient to have a strong effect on several factors. Altogether, I would be in favor of accepting this article.

Rebuttal

Reviewer 1

1. *The authors call the hybridoma cell 3A9 in the results and 389 in the letter to the referees. Then in Results and legends, they describe that the 3A9 hybridoma clone was activated for 10 minutes. However, in the letter to referees they mention that cells were activated for 5 minutes for the mass-spec experiments.* We have corrected the typo.
2. *Figure 1C should contain more detailed RNA expression kinetics; three points are insufficient.* We do not agree that 3 time points are insufficient and no explanation is provided by the Reviewer. Time points at 0, 24h, 48h clearly show the induction of RNA levels following TCR stimulation, which is the straightforward conclusion drawn from this experiment.
3. *The figure is still missing a cytosolic fraction control.* This is just not correct, we have included LAT as control of membrane vs non-membrane (or cytosolic) fractionation purity. Purpose of this experiment was to show inducible association of Rasal1 with membrane, which is clearly demonstrated by the given data. Detailed study of Rasal1 in cytosol, nucleus or other organelles is irrelevant here.
4. *The authors claim in the letter to referees and in the revised Results that PLA cannot be used to determine location of interacting proteins with no references. However, the description of the Duolink PLA system on the SIGMA website claims that this assay can be used for localization. The authors should substantiate their claims. Additionally, they state that there's no signal when anti-zeta antibodies are used with anti-Rasal1, but there is also no positive control demonstrating that any PLA assay would work with this anti-zeta reagent.* We cannot agree that the PLA system can be used to determine cellular localization. It appears that the Reviewer has not actually used the system since he/she is quoting from the SIGMA website. It is simply not correct that one can use this assay to determine the cellular localization of the protein membrane without 3D iteration. We have added images from reconstituted 293T and Jurkat cells. The zeta antibodies work in PLA, WB, flow cytometry and immuno-fluorescence without any issues.
4. *In Figures 1 and 2, authors describe that TCR-Rasal1 (Fig. 1) or ZAP70-Rasal1 (Fig. 2) interaction is CD3-dependent. But then in Figure 3 they evaluate ZAP-70 Rasal1 interaction in 293T cells which does not express TCR/CD3. They should comment on this.* The figures clearly direct an association of ZAP-70 with Rasal1. Further, we have mapped the site of binding to the kinase domain of ZAP-70. These results suggest that TCR stimulation induce necessary conformational changes in ZAP-70 in T cells allowing ZAP70 binding to Rasal1.
5. *In the description of Figure 2C in page 8, authors mislabel the lanes.* Thank you for pointing this. This has been corrected
6. *The newly included Figure 3 has many issues. The methods are poorly explained: is the FLAG-LAT used for the assay IP-ed from 293T lysates? There is no evidence that the ZAP fragments are properly folded; do the SH2 domains bind phospho-zeta? They are also missing several control blots. For example: 3B should include Rasal IP blot, 3C should include ZAP-70 and LAT blots, 3E should include a ZAP-70 blot and 3F should include a Rasal1 blot. These are standard controls that should be included.* The issue of proper folding is an unreasonable issue which raises concerns about the Reviewer's fairness. Certain ZAP-70 fragments bind to Rasal1

and therefore show specificity in binding. It is unreasonable to expect every paper to determine the conformation of each protein fragment by structural analysis. The functionality of each fragment is clearly shown by the binding of Rasal1 to the ZAP-70 protein fragments and that the loss of the kinase domain abrogates binding.

7. *Further on Fig. 3, standard curves of ZAP-70 phosphorylations should be shown for autophosphorylation and for substrate phosphorylation. Only then can the effects of Rasal1 be understood. As is, the curves in Fig 3C look odd with a dramatic effect in phosphorylation with a 2x increase in Rasal1. Coomassie is misspelled. "GST-Bora had not (sic) effect." (p.9). The decrease in pLAT in 3F is unconvincing and should be quantified. The mechanism of Rasal1 remains unclear.* The binding clearly shows altered activity as shown by alterations in the phosphorylation of LAT. The mechanism by which this occurs is logically due to the demonstrated binding of Rasal1 to the kinase domain of ZAP-70.

8. *The discussion regarding Erk activation is unclear. What's the hypothesis? That Ras is directly inhibited as Rasal1 is a GAP? That Erk is inhibited because LAT is inhibited? Because ZAP-70 is inhibited? They do some Erk assays and a mediocre Raf and Mek assay. More is needed. Fig. 4C should include total Raf and Mek blots.* The hypothesis should be obvious: Rasal1 regulates p21^{ras} which operates upstream of ERK. This should be obvious from numerous previous studies from many labs showing that p21^{ras} regulates ERK. "a mediocre Raf assay". This is a puzzling thing to say, mediocre Raf or Mek assay.. why? These comments seem to show a biased attitude of the Reviewer.

We thank the Reviewer for his/her comments and hope that the paper is now acceptable for publication.

Reviewer #3

In the revised manuscript by Thaker et al., the authors have adequately addressed the comments raised by the reviewer. We thank the Reviewer for his/her comment.

In the description of Figure 2B on page 7 and 8, lane numbers seem different from the data shown in Figure 2B. This has now been corrected.

We thank the Reviewer for his/her comments and hope that the paper is now acceptable for publication.

Reviewer #4 (Remarks to the Author):

The authors described a new interaction and function of the RASAL1 protein. They established that RASAL1 could interact with ZAP70, more precisely on the kinase domain of this protein. They showed by overexpression or by knockdown that RASAL1 is an

inhibitor of ZAP70, capable of regulating T cell proliferation and function in vitro of in vivo. These results are new and interesting. The article is convincing. We thank the Reviewer for his/her comment.

Concerning the answers to reviewer 2, the major points raised were answered to, either by new experiments or by additions in the text. We thank the Reviewer for his/her comment

The only point left unanswered is the use of CRISPR to knock out the gene more efficiently. However, the comments from the authors make sense, as only a partial knockdown or the protein is sufficient to have a strong effect on several factors. Altogether, I would be in favor of accepting this article. We thank the Reviewer for his/her comment.

We thank the Reviewer for his/her comments and hope that the paper is now acceptable for publication.

Reviewers' comments:

Reviewer #1 (Remarks to the Author):

Thaker et al.

Third review

The authors' discovery of the association of Rasal1 with the protein tyrosine kinase ZAP-1 is interesting and their studies on the functional effects of this association are provocative. This interaction might someday be exploited clinically. However, the manuscript suffers because the mechanistic studies and interpretation of the results are inadequate.

- The text on the top of p.7 provides an incomplete conclusion of the Fig 2 data. I conclude that a fraction of Rasal1 associates with a fraction of ZAP-70 (to amend 2nd to last sentence of the section), and that Rasal1 associates with the ZAP-70 that is recruited to the TCR upon activation (to modify last sentence).

- Figure 3 remains incomplete, as several of the control blots that I requested are still lacking. More important is that the mechanism of the Rasal1 inhibition of ZAP-70 is not clear. Does an enzymatically dead form of Rasal1 inhibit? The authors did a structure-function analysis of ZAP-70 to identify that the kinase domain was the target of Rasal1. Why not present a similar analysis of Rasal1? I realize that this latter experiment was not proposed in earlier reviews, but I did previously conclude, that "the mechanism of Rasal1 (inhibition) remains unclear." It still is.

- Similarly, the mechanism of Rasal1 inhibition of Erk is unclear, as previously stated. The authors think that the explanation "should be obvious. Rasal1 regulates p21ras which operates upstream of Erk." That might be the explanation, but the previous section of the manuscript demonstrated that Rasal1 inhibited ZAP-70. Certainly, inhibition of ZAP-70 would also result in lack of Erk activation, or, as is potentially mis-described, "inhibition" by blocking tyrosine phosphorylation of the adaptors and enzymes necessary to activate Ras and thus Ras-dependent effects. Results in Fig4C (not described in the text), the inhibition of Raf and Mek activation, though inadequately developed, support this interpretation. The authors' response to a previous request to expand analysis of this issue is not adequate.

Rebuttal

Reviewer 1

(1) We have added the word “a portion of” to the text ““that a portion of Rasal1 associates with ZAP-70 of the TCR complex where Rasal1 associates with the ZAP-70 and is recruited to the TCR upon activation” (Pg. 7, first para, lines 5-6).

(2) We have included control blots: Rasal1 control blot (3B), ZAP-70 and LAT blots (3C), a ZAP-70 control blot (3E) and a quantification histogram and Rasal1 blot (3F).

(3) Lastly, from discussions with the Editor, it was agreed that further experiments involving the mapping of binding sites in Rasal1 are outside the scope of the paper. Instead, as requested by the Editor, we have provided a statement outlining the limitations of the study, “There are therefore two pathways by which Rasal1 negatively influences T-cell activation underscoring its potential importance. The degree to which the inhibition of ZAP-70 versus the p21^{ras} pathway regulate different aspects of T-cell activation and functions remains to be determined in future studies. ZAP-70 signaling has been linked to altered thymic differentiation, cytoskeletal remodeling and certain effector functions⁶¹, while p21^{ras} has been reported to regulate T-cell non-responsiveness or anergy²⁰” (Pg. 15, 1st para, lines 12-18). We thank the Reviewer for his/her helpful suggestions.